# Testosterone-induced metabolic changes in seminal vesicle epithelium modify seminal plasma components with potential to improve sperm motility

**Takahiro Yamanaka[1], Zimo Xiao[1], Natsumi Tsujita[1], Mahmoud Awad[1,2], Takashi Umehara[1], Masayuki Shimada[1,3]\***

[1]Graduate School of Integrated Sciences for Life, Hiroshima University, Higashihiroshima, Hiroshima, Japan; [2]Department of Histology, Faculty of Veterinary Medicine, South Valley University, Qena, Egypt; [3]Graduate School of Innovation and Practice for Smart Society, Hiroshima University, Higashihiroshima, Hiroshima, Japan

**\*For correspondence:**
mashimad@hiroshima-u.ac.jp

## eLife Assessment

This **important** work elucidates the biological processes and detailed mechanisms by which testosterone influences seminal plasma metabolites in mice. The evidence supporting the upregulation of metabolic enzymes and the role of ACLY is **solid**, highlighting the potential contributions of fatty acids to sperm motility.

## Abstract

Male factors account for almost half of the causes of infertility. In rodents and humans, most of the components of semen are supplied by the seminal vesicles, and they support male reproductive ability, but there are many unknown details. This study focused on metabolic changes in seminal vesicle epithelial cells and investigated how testosterone affects seminal plasma composition. A factor improving the linear motility of sperm was secreted from the seminal vesicles, and it was produced in an androgen-dependent manner. Bioassays, gene expression, and flux analysis studies demonstrated that testosterone promotes glucose uptake in seminal vesicle epithelial cells via GLUT4, resulting in fatty acid synthesis. ACLY was a critical factor in this metabolic change, which produced fatty acids, especially oleic acid. In conclusion, the critical role of testosterone-induced metabolic changes in the seminal vesicles is to ensure the synthesis of fatty acids. These findings suggest that testosterone-dependent lipid remodeling may contribute to sperm straight-line motility, and further functional verification is required.

## Introduction

Infertility is defined by the World Health Organization (WHO) as the failure to conceive within 12 months despite contraceptive-free sexual intercourse. Its prevalence has increased over the decades and is observed in 17.5% of couples worldwide (*World Health Organization, 2023*). The male partner is thought to be solely or in combination responsible for 50% of infertility cases, and the male factor in the etiology of infertility is highly recognized. Male fertility depends largely on the quality of semen composed of sperm and seminal plasma and decreases with aging (*Kidd et al., 2001*; *Mumcu et al., 2020*; *Xu et al., 2020*).

Sperm that have completed spermatogenesis in the testis acquire the potential to fertilize during maturation in the epididymis (*Elbashir et al., 2021*; *Cosentino and Cockett, 1986*; *Howarth, 1983*).

The physiological changes of sperm during the fertilization process are collectively referred to as 'capacitation'. This change is characterized by large-amplitude flagellar beating (called hyperactivation) and developing the capacity to undergo the acrosome reaction and can be induced by culturing sperm collected from the epididymis in a medium containing bicarbonate and cholesterol acceptors (*Yanagimachi, 1970*; *Yanagimachi, 1994*). However, once capacitation is complete, sperm cannot maintain this state for a long time. Therefore, even if epididymal sperm that have not been exposed to seminal plasma are artificially inseminated into the cervix or uterus, the fertilization rate remains low (*Gaddum-Rosse, 1981*; *Shalgi et al., 1992*; *Li et al., 2015*). This is because, in vivo, during ejaculation, exposure of epididymal sperm to seminal plasma masks the unintended capacitation as they pass through the female reproductive tract and ensures fertilization of sperm that reaches the oviduct (*Noda and Ikawa, 2019*). In other words, seminal plasma plays an important role in fine-tuning the timing of sperm capacitation and in maintaining the sustained sperm motility required to reach the oviduct.

Seminal plasma is mainly produced in the seminal vesicles (approximately 65–75%), with approximately 20–30% produced by the prostate, and small amounts secreted by the bulbourethral and urethral glands (*Verze et al., 2016*). The mixing of seminal fluid with accessory reproductive gland fluid during ejaculation causes semen to clot and liquefy in humans and causes vaginal plug formation in rodents. Vaginal plugs not only inhibit continuous mating but also play a role in maintaining sperm in the uterus and ensuring male fertility (*Noda and Ikawa, 2019*). TGFβ, a component of seminal plasma, increases antigen-specific Treg cells in the uterus of mice and humans, which induces immune tolerance, resulting in pregnancy (*Balandya et al., 2012*; *Shima et al., 2015*). Furthermore, SVS2, secreted from the seminal vesicle, is also known to be required for sperm to escape attack by the female immune system in the uterus (*Kawano et al., 2014*). In addition, seminal plasma contains various biochemical components that support sperm metabolism, including lipids and organic acids. Differences in the relative size of the accessory reproductive glands reflect differences in the seminal plasma composition. Accordingly, there are considerable differences in the properties of seminal plasma among animal species. Specifically, *Rosecrans et al., 1987* found significant differences in the levels of most biochemical components in seminal plasma compared to those in blood, with specific factors like fructose being rarely detected in blood but abundant in seminal plasma. Our previous studies (*Umehara et al., 2018*; *Zhu et al., 2019*; *Islam et al., 2021*) documented that seminal plasma also contains creatine and fatty acids taken up by sperm within minutes. These substances play crucial roles in regulating sperm metabolic pathways and enhancing sperm motility. These observations indicate that seminal plasma not only alters the female immune system to protect sperm but also directly influences sperm metabolic pathways, facilitating the transformation of sperm capable of in vivo fertilization.

Several specific factors produced by the male accessory glands that contribute to seminal plasma and impact male fertility have been elucidated. For example, surgical removal of seminal vesicles in male mice and rats was associated with infertility (*Kawano et al., 2014*; *Noda et al., 2019*; *Queen et al., 1981*). The observations that fructose (*Mann, 1946*) and citrate (*Humphrey and Mann, 1949*) concentrations in seminal plasma of control mice and rats are higher than in castrated animals suggest that the specific metabolism of the accessory glands might be affected by testosterone derived from the testes, which activate intracellular androgen receptors (AR; NR3C4) required for gene regulation of transcription. AR-deficient mice have been found to be infertile (*Chang et al., 2013*), and long-term administration of androgen receptor antagonists, like flutamide, abolished male fertility (*Fix et al., 2004*; *Nagaosa et al., 2007*). AR-deficient mice and transcriptome analyses have identified specific targets of AR (*Smith and Walker, 2014*; *Cooke and Walker, 2021*). In both mice and humans, androgen levels decrease with aging, and their concentrations have been reported to cause a decrease in male fertility, especially ejaculated sperm motility. In addition to impairments in spermatogenesis, a decline in androgen levels may also disrupt the metabolic functions of the accessory glands and alter the composition of seminal plasma. Therefore, the intricate interplay between testosterone and seminal plasma synthesis in the accessory glands requires further investigation.

In this study, various bioassays were done to identify which accessory glands are essential for sperm fertility. Subsequently, comparative in vivo and in vitro analyses were performed using a hypoandrogenic model. In addition, biochemical and molecular techniques such as RNA-seq, flux analysis, mass spectrometry, and knockdown experiments were employed. These analyses revealed that

testosterone promotes the synthesis of oleic acid in seminal vesicle epithelial cells and its secretion into seminal plasma.

## Results

### The effect of androgen-dependent changes in mouse seminal vesicle secretions on the linear motility of sperm

A direct comparison was conducted to clarify the effects of secretions from the prostate and seminal vesicles on the motility parameters of epididymal sperm. Since it is difficult to collect the ejaculated semen of mice, a pseudo-seminal plasma was prepared using seminal vesicle secretions and homogenized prostate tissue (*Figure 1*, *Figure 1—figure supplement 1A*). When seminal vesicle secretions were added to the human tubal fluid (HTF) medium, the sperm motility, progressive motility, the straight line velocity (VSL), and the percentage of linearity (LIN) values increased significantly compared to when only prostate extract was used. However, the curvi-linear velocity (VCL) did not change (*Figure 1B–F*). Meanwhile, the secretions from the prostate and seminal vesicles did not affect the viscosity of the medium (*Figure 1G*). These results indicate that sperm exposure to seminal plasma factors significantly improved sperm linear motility (high VSL and LIN).

Androgens regulate the functions of seminal vesicles. Therefore, to elucidate whether the seminal vesicle factors, which enhance sperm linear motility, are synthesized in an androgen-dependent manner, bioassays were performed using seminal vesicle secretions collected from control mice or mice treated daily for 7 consecutive days with flutamide, an androgen receptor antagonist (*Figure 1H*, *Figure 1—figure supplement 1B*). The results showed that motile, progressive motile, LIN, and VSL were significantly reduced in the sperm cultured in seminal vesicle secretions from flutamide-treated (Flutamide) compared to the sperm cultured in seminal vesicle secretions from controls (Ctrl) (*Figure 1I–L*). The reductions in VSL, motile, and progressive motile were also significantly lower in the sperm treated with seminal vesicle secretions from mice over 12 months of age that showed low circulating testosterone levels as compared with the sperm treated with seminal vesicle secretions from young adult mice (3-month old mice) (*Figure 1—figure supplement 2A, B*). Furthermore, JC-1 staining to measure mitochondrial membrane potential, which is important for sperm linear motility, showed a significantly lower percentage of high mitochondrial membrane potential in sperm treated with the secretions of seminal vesicles from flutamide-treated mice and mice over 12 months of age (*Figure 1M*, *Figure 1—figure supplement 1C, D*, *Figure 1—figure supplement 2C*). Since flutamide treatment and aging also affect spermatogenesis and sperm motility, it should be noted that epididymal sperm donors differ from seminal vesicle donors (*Figure 1H*). Due to the strong effect of the decapacitation factor, it was found that seminal vesicle fluid, regardless of the presence or absence of androgens, reduces the fertilization rate in in vitro fertilization (IVF; *Figure 1—figure supplement 1E*). The baseline information for the epididymal sperm used in the experiment showed that the sperm were normal (*Figure 1—figure supplement 1A*). These results indicated that androgen-dependent changes in seminal vesicle functions alter the levels/activity of seminal vesicle secretion factors, which regulate sperm linear motility. The changes in seminal vesicle functions were also induced by aging.

### Testosterone–androgen receptor activity suppresses the proliferation of seminal vesicle epithelial cells

To understand the difference between seminal vesicles that secrete factors for enhancing sperm linear motility and those that are impaired in this function, we performed histological analysis of seminal vesicles recovered from mice treated and non-treated with flutamide. In control mice, the seminal vesicle epithelial cells were in a single layer, and androgen receptors were localized in the nuclei of seminal vesicle epithelial cells (*Figure 2A*). However, in mice treated with continuous administration of flutamide, seminal vesicle epithelial cells were multilayered, and the strong accumulation of androgen receptors in the nucleus was not observed (*Figure 2A*). To investigate the cause of this multilayering of cells, we performed Ki67 staining, a marker of cell proliferation, and apoptosis detection by TUNEL staining (*Figure 2B*). The results showed that the percentage of Ki67-positive cells increased significantly after treatment with flutamide, but the percentage of TUNEL-positive cells did not change, suggesting that the testosterone–androgen receptor pathway is responsible for the inhibition of proliferation of seminal vesicle epithelial cells (*Figure 2C*). These changes of abnormal morphology

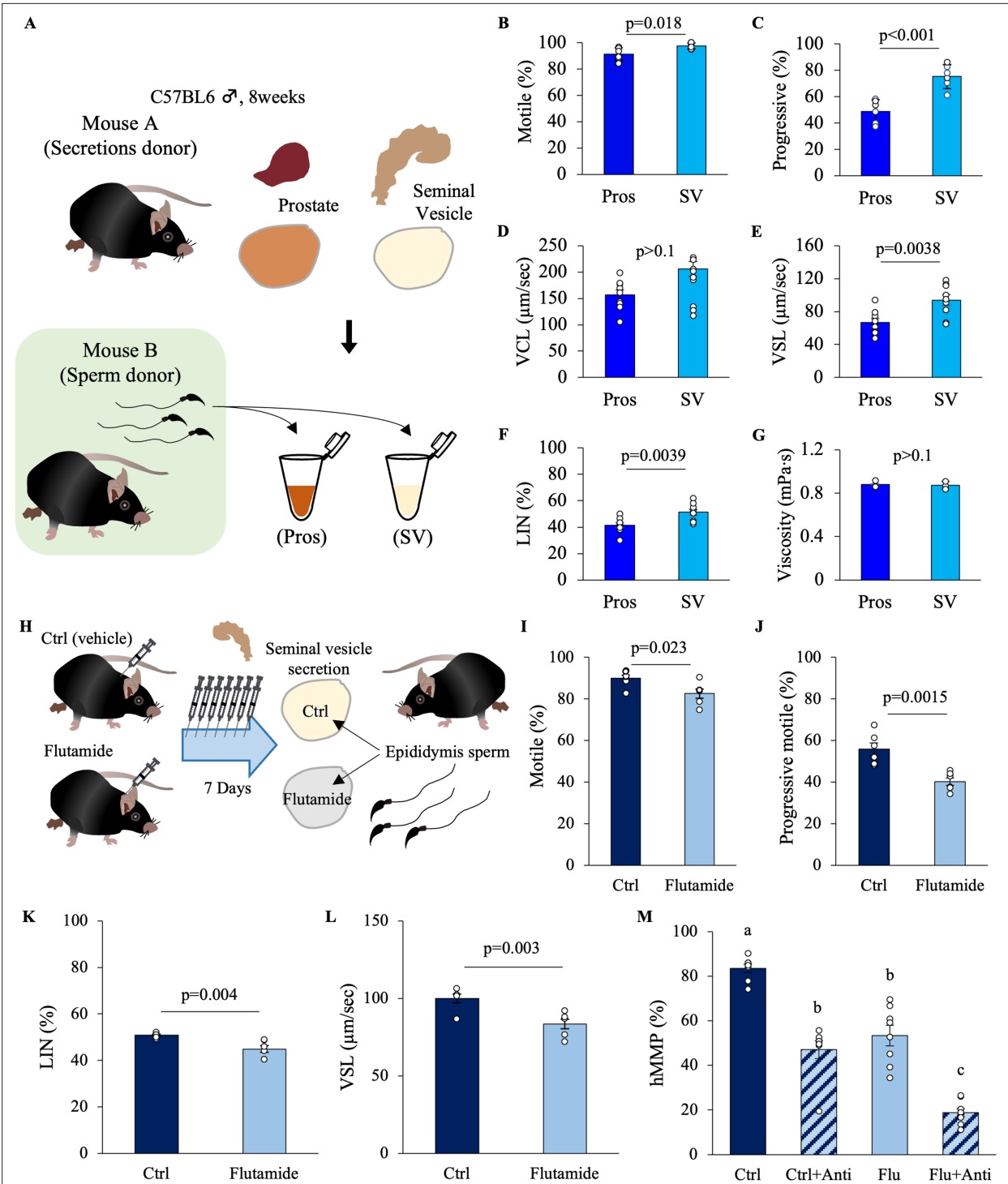

**Figure 1.** Effects of seminal vesicle fluid and prostate fluids on sperm motility. (**A**) The effects of secretions from the prostate and seminal vesicles on sperm motility were directly compared. The effects of secretions from the prostate (Pros) or seminal vesicles (SV) on the motility parameters of epididymal sperm were tested: (**B**) motile, (**C**) progressive motile, (**D**) curvi-linear velocity (VCL), (**E**) straight-line velocity (VSL), and (**F**) linearity (LIN) is the ratio of VSL to VCL of sperm that was incubated with the mixture of seminal vesicle secretions or prostate extracts. (**G**) Viscosity of a solution containing prostate or seminal vesicle secretion with a protein concentration of 10 mg/ml. (**H**) Experimental design to evaluate the effect of seminal vesicle secretions collected from male mice treated with or without flutamide (50 mg/kg subcutaneously every day for 7 days) on sperm. Note that the donor mice for sperm and seminal vesicle secretions were different. Performed bioassay using seminal vesicle fluid after treatment with vehicle (Ctrl) or flutamide: (**I**) motile, (**J**) progressive motile, (**K**) LIN, and (**L**) VSL. (**M**) The high mitochondrial membrane potential (hMMP) was checked by the JC-1 kit. Antimycin (Anti) was used as the negative control for hMMP. Data are mean ± SEM. *n* = 3–9 independent replicates. The viscosity measurement

*Figure 1 continued on next page*

*Figure 1 continued*

data were from *n* = 3 repeated experiments using pooled prostate or seminal vesicle extracts from at least 20 mice. Percentage data were subjected to arcsine transformation before statistical analysis. (**B–G, I–L**) Significance was tested in comparison using Student's *t*-test. (**M**) Since the results of the Bartlett test were significant, a two-way ANOVA using a generalized linear model was performed. Both the flutamide administration and antimycin addition were significant. The interaction was not significant. Games–Howell was performed as a post hoc test. Different letters represent significantly different groups. Data were considered statistically significant at p < 0.05.

The online version of this article includes the following source data and figure supplement(s) for figure 1:

**Source data 1.** Raw data and sample metadata for *Figure 1*.

**Figure supplement 1.** Sperm quality control data and exploratory research into the effects of seminal vesicle secretions on sperm fertilization.

**Figure supplement 1—source data 1.** Raw data and sample metadata for *Figure 1—figure supplement 1*.

**Figure supplement 2.** Characteristics of the seminal vesicle in aging mice.

**Figure supplement 2—source data 1.** Raw data and sample metadata for *Figure 1—figure supplement 2*.

**Figure supplement 3.** Gating strategy of flow cytometry for sperm.

and high proliferation activity were also observed in seminal vesicles of mice older than 12 months that showed low serum testosterone levels (*Figure 1—figure supplement 2D–F*). Therefore, seminal vesicle epithelial cells were collected and cultured in primary culture, as shown in *Figure 2D*. The changes in cell number were counted, and the cell cycle was analyzed by flow cytometry. When cultured without testosterone (c), the number of seminal vesicle epithelial cells increased in a time-dependent manner. However, the addition of testosterone significantly decreased the number of seminal vesicle epithelial cells during the 8-day culture period in a concentration-dependent manner (*Figure 2E*). Flow cytometry on day 7 revealed a significant increase in the proportion of cells in the G0/G1 phase and a corresponding decrease in the G2/M phase. The G2/M phase was significantly decreased by the addition of testosterone (*Figure 2F, G*). These results indicate that the testosterone-androgen receptor pathway inhibits cell division in seminal vesicle epithelial cells.

## Testosterone alters metabolic gene expression in seminal vesicle epithelial cells

To better understand the roles of testosterone in suppressing the proliferation of seminal vesicle epithelial cells, RNA sequencing was done using the epithelial cells cultured with or without testosterone. DEG (differentially expressed genes) analysis revealed that a total of 23,459 genes were identified, including 997 upregulated genes and 3463 downregulated genes in seminal vesicle epithelial cells by testosterone relative to control (*Figure 3A*). Genes largely upregulated by testosterone included those already reported in seminal vesicles (*Kawano et al., 2014*; *Noda et al., 2019*; *Smyth et al., 2022*), such as the Seminal Vesicle Secretion (*Svs*) family (*Svs1–6*), *Pate4*, *Spinkl*, *Ceacam10*, and *Sva* (*Figure 3B, C*). Therefore, the primary culture of seminal vesicle epithelial cells used in this experiment was considered to maintain the characteristics of seminal vesicle epithelial cells in vivo. Many genes were downregulated by testosterone, and those with a high rate of variation included collagen synthase and *Mmp2*, which are involved in extracellular matrix formation (*Figure 3C*). However, KEGG analysis, rather than individual genes, was used to predict functions altered by testosterone. It was found that the expression of genes involved in the metabolic pathway was the most highly affected by testosterone (*Figure 3D*). Because there is a close relationship between cell proliferation and metabolic activity and between metabolic substrates in seminal plasma and sperm motility, we hypothesized that testosterone-induced changes in seminal vesicle epithelial cell metabolism alter the components of seminal plasma to mediate the enhancement of sperm motility.

## Testosterone shifts the metabolic flux of seminal vesicle epithelial cells

In the presence of testosterone, the addition of glucose significantly activated the extracellular acidification rate (ECAR) normalized by the total number of live cells compared to that in epithelial cells cultured without testosterone (*Figure 4A–C*). These data indicate that testosterone enhances glucose utilization. This leads to the interpretation that testosterone increases the flow of glycolysis by increasing glucose uptake and alters metabolic flux distribution. To determine the metabolic fate of glycolytic end products, seminal vesicle epithelial cells were cultured with and without testosterone. Subsequently, the concentrations of pyruvate and lactate in the culture supernatant normalized by

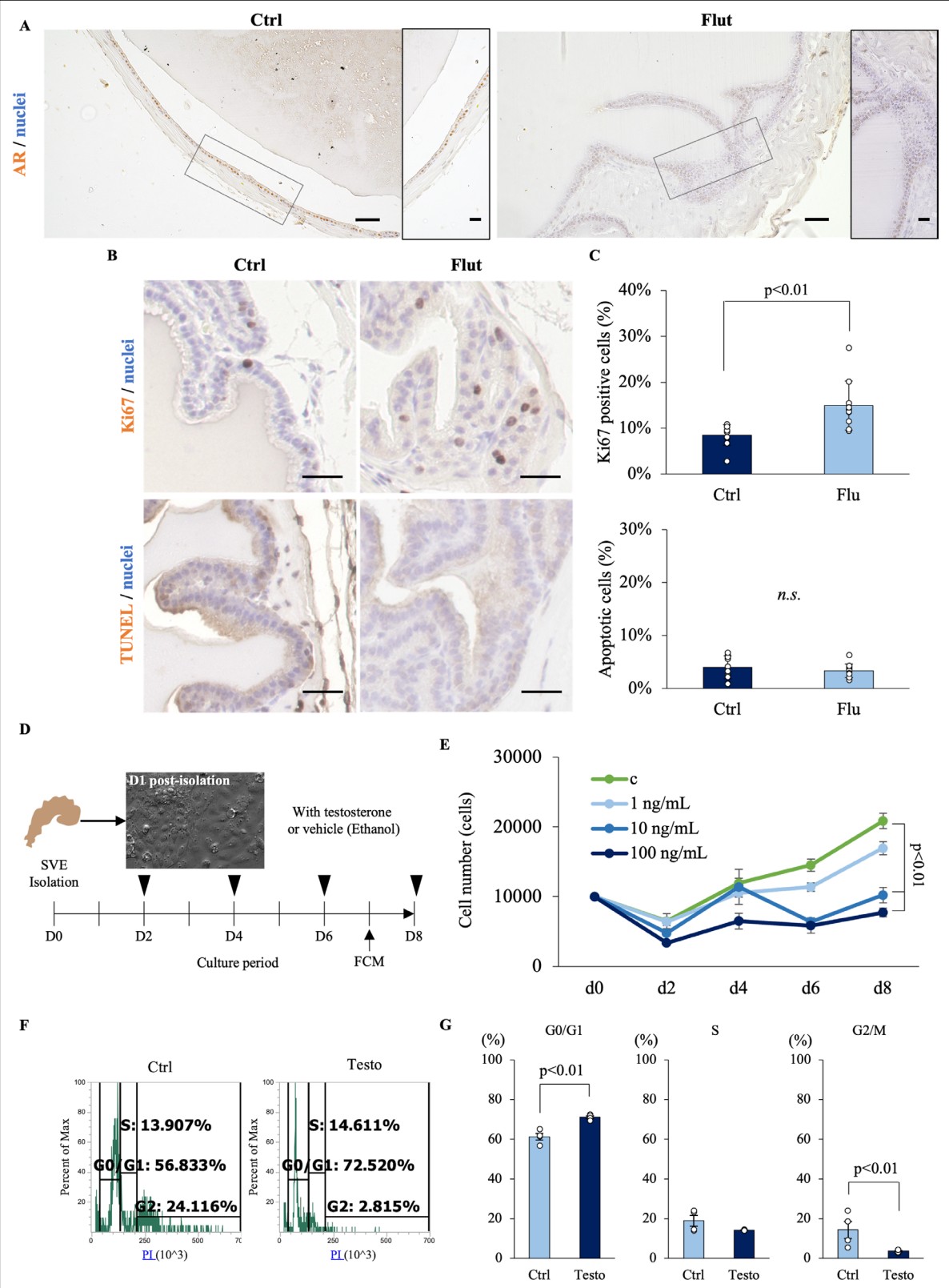

**Figure 2.** The testosterone–androgen receptor pathway inhibits the cell proliferation of seminal vesicle epithelial cells. (**A**) The localization of the androgen receptor (AR; NR3C4) in seminal vesicle of control (Ctrl) mice and those treated with flutamide (Flut). Scale bar = 50 μm from ×10 magnification for left panel and 20 μm from ×20 magnification for right panel in each group. (**B, C**) Cell proliferation and apoptosis in seminal vesicle of control (Ctrl) mice and those treated with flutamide (Flut): (**B**) Representative images of Ctrl and Flut staining for Ki67 (top) and TUNEL (bottom). Scale

*Figure 2 continued*

bar = 20 μm. (**C**) Percentage of Ki67-positive cells (top) and TUNEL-positive (bottom) in Ctrl and Flut-treated seminal vesicle sections. (**D**) Experimental design to evaluate the effects of testosterone on the proliferation of seminal vesicle epithelial cells in vitro. The epithelial cells were cultured with or without testosterone for 8 days. (**E**) Growth curves of seminal vesicle epithelial cells that were cultured with 0 (c), 1, 10, and 100 ng/ml of testosterone. (**F, G**) Cell cycle status was determined by flow cytometry. Data are mean ± SEM. $n$ = 3–5 mice or independent replicates. Each replicate experiments with 3–6 wells containing pooled cells from 3 to 5 mice. Percentage data were subjected to arcsine transformation before statistical analysis. Significance was tested in comparison to Ctrl using Student's $t$-test. The cell number data at d8 was compared with the control using Dunnett's test. Data were considered statistically significant at $p < 0.05$.

The online version of this article includes the following source data and figure supplement(s) for figure 2:

**Source data 1.** Raw data and sample metadata for *Figure 2*.

**Figure supplement 1.** Gating strategy of flow cytometry for seminal vesicle epithelial cells.

the total number of live cells were measured. The addition of testosterone significantly decreased the pyruvate concentration; however, the lactate concentration did not change significantly (*Figure 4D, E*). We therefore examined cellular energy metabolism with a flux analyzer, anticipating that testosterone would elevate glycolytic flux, thereby producing more pyruvate from phosphoenolpyruvate. Because extracellular pyruvate levels simultaneously declined, we inferred that the cells had an increased pyruvate demand and, at that time, hypothesized that the excess pyruvate would enter the mitochondria to support enhanced oxidative metabolism. Unexpectedly, when measuring mitochondrial respiratory metabolism normalized by the total number of live cells with the flux analyzer, the basal respiratory rate (the electron transport chain activity) was significantly decreased by testosterone (*Figure 4F–J*). Oligomycin-sensitive respiration was also significantly suppressed by exposure to testosterone. Furthermore, the FCCP uncoupled respiration and spare respiration capacity were also significantly decreased in response to testosterone treatment. Overall, these data indicate that testosterone promotes the redistribution of metabolic flux. In other words, testosterone increased glycolysis in seminal vesicle epithelial cells while decreasing mitochondrial respiration. To determine whether these changes were accompanied by changes in gene expression of specific metabolic-related enzymes, we analyzed gene expression levels.

## Testosterone causes changes in the expression of genes encoding enzymes involved in glucose catabolism and anabolism

The expression of genes involved in glucose catabolism and anabolism in the in vitro culture system of seminal vesicle epithelial cells was analyzed by qPCR (*Figure 5*). The gene expression of glucose transporter (*Slc2a*) and enzymes involved in the glycolytic pathway was not significantly affected by testosterone, consistent with the results of metabolic flux analysis, which showed that the glycolytic pathway was not enhanced despite the activation of glycolysis. Furthermore, testosterone did not significantly affect the expression of genes encoding enzymes involved in the tricarboxylic acid (TCA) cycle in mitochondria. However, the expression of genes encoding enzymes involved in the electron transport chain was significantly decreased. Interestingly, testosterone significantly increased the expression of *Acly*, which encodes a cytoplasmic enzyme that converts citrate transported from the TCA cycle into acetyl-CoA, a substrate that is essential for fatty acid synthesis. Although *Hmgcr*, which encodes the rate-limiting enzyme involved in cholesterol synthesis, was unchanged, the expression of *Acc*, which encodes the rate-limiting enzyme (ACC1; ACACA) for fatty acid synthesis, was significantly upregulated by testosterone. These characteristic testosterone-induced changes in gene expression observed in cultured cells were also detected by ACLY immunostaining of in vivo seminal vesicle samples (*Figure 5—figure supplement 1*). Specifically, the expression of genes encoding a group of enzymes of the electron transport chain (*mtNd6*) was significantly upregulated by continuous flutamide administration in vivo, whereas the gene expression of *Acly* and *Acc* was significantly decreased by flutamide administration (*Figure 5—figure supplement 1B*). Immunostaining results showed that ACLY was detected specifically in seminal vesicle epithelial cells, and their staining appeared to be reduced by flutamide treatment. The lower level of ACLY was also observed in seminal vesicle epithelial cells of mice over 12 months of age (*Figure 5—figure supplement 1C*).

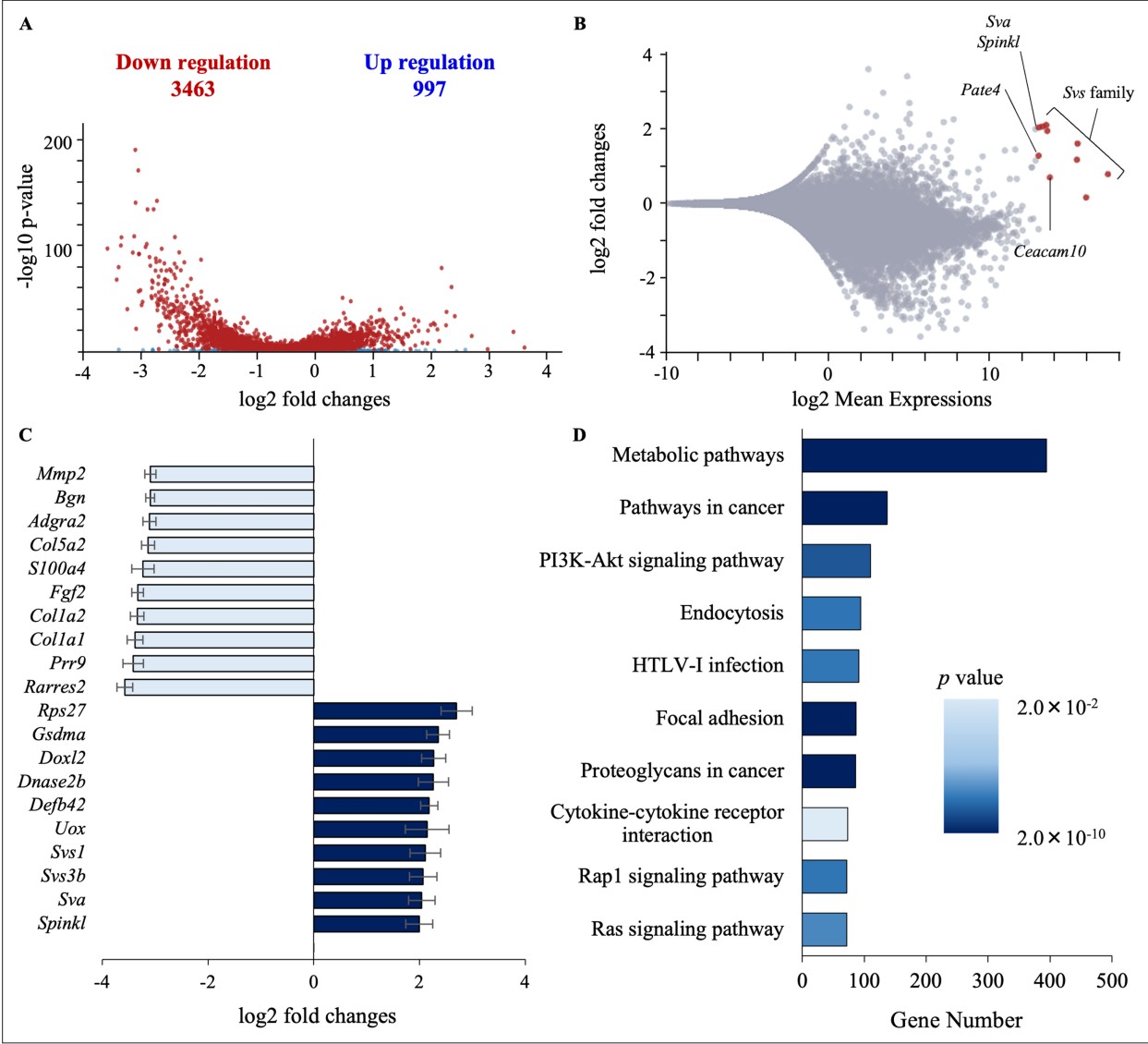

**Figure 3.** Testosterone changes the expression of genes involved in metabolic pathway in seminal vesicle epithelial cells. (**A**) Volcano plot of differentially expressed genes. RNA sequencing was performed using RNA extracted from the seminal vesicle epithelial cells cultured with or without 100 ng/ml testosterone. Genes with a significant expression change are highlighted as red dots. A total of 4460 genes were significant for a cutoff of p < 0.05. (**B**) MA plot of differentially expressed results. The seminal vesicle-specific genes are highlighted as red dots rather than significant genes. (**C**) Top 10 genes upregulated or downregulated by testosterone, respectively, for altered gene expression. Data are mean fold changes ± SEM. $n = 3$ independent replicates. (**D**) Conducted KEGG analysis to identify differences in pathway enrichment by testosterone, identifying the most variability in metabolic pathway genes. It shows the number of differentially expressed genes annotated to Gene Ontology shows (x-axis). p-value (adjusted) which measures the statistical significance of a possible functional enrichment for each term.

The online version of this article includes the following source data for figure 3:

**Source data 1.** Raw data and sample metadata for *Figure 3*.

## Mechanism of testosterone-induced enhancement of glucose uptake and, thereby, fatty acid synthesis

Since it is known that testosterone regulates the intracellular localization of GLUT4, we focused on the intracellular localization of GLUT4 and its glucose uptake capacity (*Murata et al., 2002*; *Wilson et al., 2013*). The effect of testosterone on glucose uptake in seminal vesicle epithelial cells was analyzed using fluorescence-labeled glucose. As expected from the results of flux metabolic analyses, testosterone treatment significantly increased glucose uptake (*Figure 6A*). Indinavir, a functional inhibitor of GLUT4, significantly suppressed glucose uptake and significantly decreased fatty acid synthesis

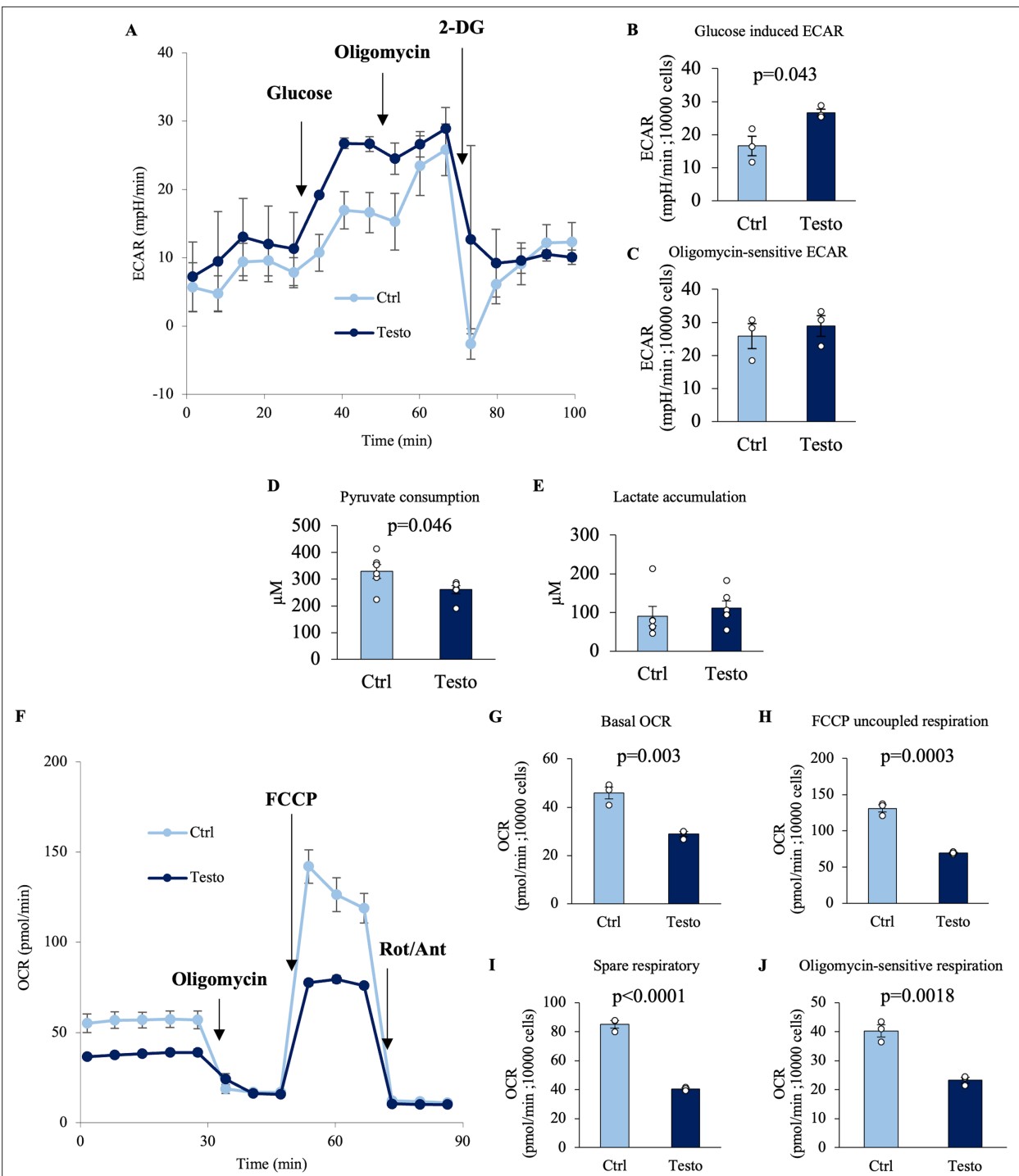

**Figure 4.** Testosterone regulates glucose metabolism and mitochondrial ATP production in seminal vesicle epithelial cells. (**A–C**) Extracellular acidification rate (ECAR) analysis by an extracellular flux analyzer in seminal vesicle epithelial cells cultured with 100 ng/ml testosterone (Testo) or vehicle (Ctrl) for 7 days: (**A**) ECAR kinetics of seminal vesicle epithelial cells using an extracellular flux analyzer. (**B**) Glucose induced ECAR and (**C**) oligomycin-sensitive ECAR. (**D, E**) The concentrations of pyruvate and lactate in the culture supernatant. These measurements were normalized based on cell count and viability. (**D**) Pyruvate concentration in the medium where seminal vesicle epithelial cells were cultured with or without testosterone for 24 hr. (**E**) Lactate concentration. (**F–J**) Mitochondrial respiration measurement by an extracellular flux analyzer: (**F**) Oxygen consumption rate (OCR) kinetics of seminal vesicle epithelial cells with or without 100 ng/ml testosterone. (**G**) Basal OCR, (**H**) FCCP uncoupled respiration, (**I**) spare respiratory capacity, and (**J**) oligomycin-sensitive respiration. Data are mean ± SEM of $n$ = 3 replicate experiments with 3–6 wells containing pooled cells from 3 to 5 mice or medium. Student's $t$-test was used to compare Ctrl and Testo. The cells were normalized to 10,000 viable cells/well immediately before analysis. Data were considered statistically significant at p < 0.05. The viability of the cells before experiments was 86–93%.

*Figure 4 continued on next page*

*Figure 4 continued*

The online version of this article includes the following source data for figure 4:

**Source data 1.** Raw data and sample metadata for *Figure 4*.

(the amount of fatty acids accumulated in the cell), which was increased by testosterone (*Figure 6A, B*). Furthermore, testosterone treatment induced translocation of GLUT4 to the plasma membrane of seminal vesicle epithelial cells (*Figure 6C*). These data suggest that GLUT4 was partially localized at the plasma membrane and contributed to glucose uptake. In contrast, in seminal vesicles from mice continuously treated with flutamide, no strong signal was detected at the presumed location of GLUT4 near the plasma membrane of seminal vesicle epithelial cells, even though the protein expression level was not affected (*Figure 6C, D*). The same characteristics were observed in aged mice.

Qualitative and quantitative analysis of fatty acid synthesis was performed by gas chromatography–mass spectrometry (GC–MS). In seminal vesicles, C16:0, C18:0, and C18:1 were detected, in which C18:1 was decreased to an undetectable level in seminal vesicles of mice treated with the flutamide (*Figure 6—figure supplement 1A*). In the cultured seminal vesicle epithelial cells, C18:1, oleic acid, was significantly increased by the addition of testosterone as observed in the seminal vesicle (*Figure 6E*). The expression of *Elovl6* encoding ELOVL6, which elongates C16:0 to C18:0, was significantly up-regulated by testosterone (*Figure 6F, G*). The testosterone-induced change was also detected in in vivo samples (seminal vesicle), where the expression of *Elovl6* was decreased by the flutamide treatment (*Figure 6—figure supplement 1B*).

In order to examine whether these lipids are the factors that induce sperm linear motility, we examined whether or not the addition of lipid mixture (LM) changed the sperm motility pattern (*Figure 6—figure supplement 2*). The LM did not affect the overall percentage of motile or progressively motile

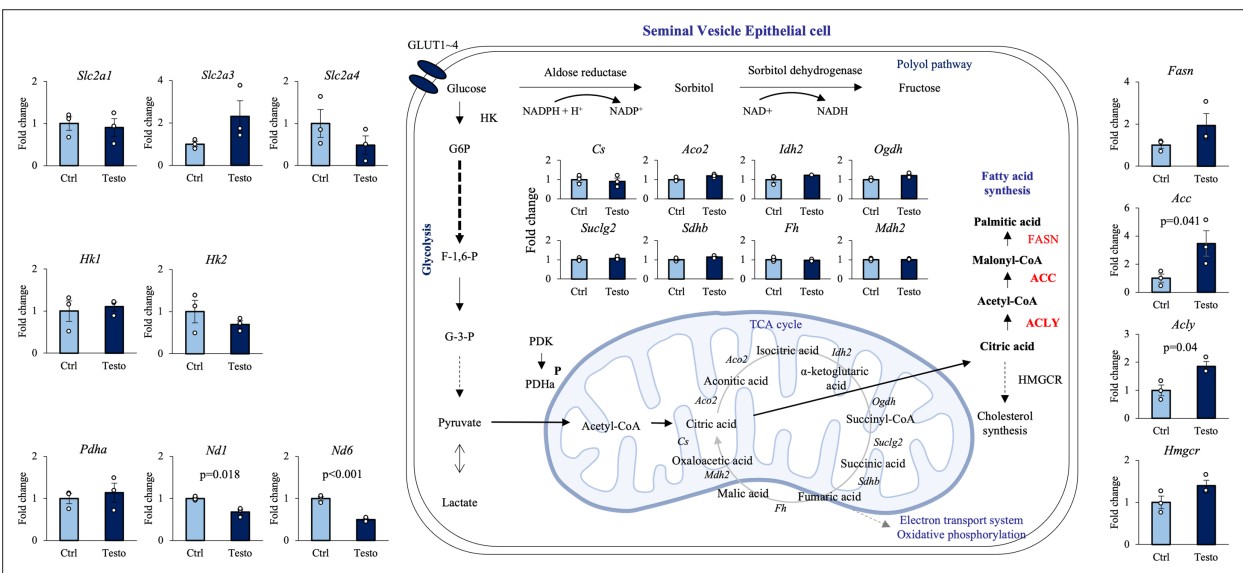

**Figure 5.** Effect of testosterone on gene expression of enzymes involved in the glucose metabolic pathway in seminal vesicle epithelial cells. To elucidate the effects of testosterone on gene expression of enzymes involved in glucose catabolism and anabolism in seminal vesicle epithelial cells. ctrl: 7 days of culture with vehicle. Testo: 7 days of culture with 100 ng/ml testosterone. Data are mean ± SEM of *n* = 3 replicate experiments with 3–6 wells containing pooled cells from 3 to 5 mice. Student's *t*-test was used to compare Ctrl and Testo. Data were considered statistically significant at p < 0.05.

The online version of this article includes the following source data and figure supplement(s) for figure 5:

**Source data 1.** Raw data and sample metadata for *Figure 5*.

**Figure supplement 1.** ACLY expression in in vivo seminal vesicle.

**Figure supplement 1—source data 1.** Raw data and sample metadata for *Figure 5—figure supplement 1*.

**Figure supplement 1—source data 2.** PDF file containing original western blots for *Figure 5—figure supplement 1A*, indicating the relevant bands and treatments.

**Figure supplement 1—source data 3.** Original files for western blot analysis displayed in *Figure 5—figure supplement 1A*.

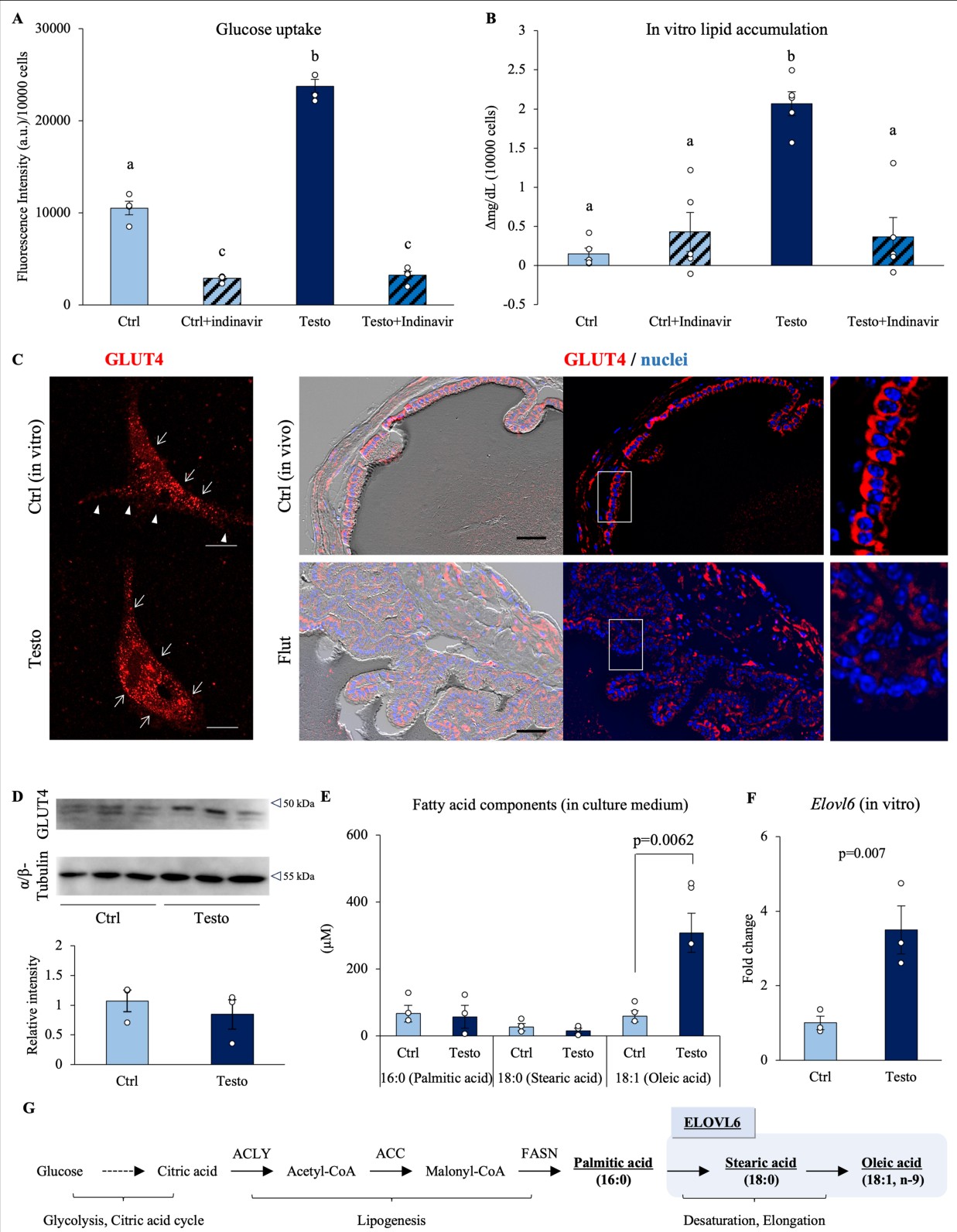

**Figure 6.** Testosterone enhances glucose uptake and fatty acid synthesis via GLUT4 trans-localization in seminal vesicle epithelial cells. (**A**) The glucose uptake ability of seminal vesicle epithelial cells was detected using fluorescence-tagged glucose. The cells were cultured with 100 ng/ml testosterone and/or indinavir (100 µM) for 7 days and then further treated with glucose uptake probe-green for 15 min, and intracellular fluorescence was measured by flow cytometry (*n* = 4). (**B**) Lipid accumulation in the medium where seminal vesicle epithelial cells were incubated for 24 hr (*n* = 6). (**C**) Left panel:

*Figure 6 continued on next page*

*Figure 6 continued*

Immunostaining of GLUT4 in cultured cell with 100 ng/ml testosterone or vehicle (in vitro model). Right panel: Immunostaining of GLUT4 in flutamide-treated or vehicle mice derived seminal vesicle (in vivo model). Representative images from the same field of view. Left, merged image (fluorescence overlaid with bright-field). Middle, fluorescence channel only. Right, magnified view of the boxed region in the left panel. Scale bar is 10 µm (left panel) or 50 µm (right panel). (**D**) Western blot images of GLUT4 and α/β-tubulin in three sets of seminal vesicle epithelial cells cultured with 100 ng/ml testosterone (Testo) or in vehicle (Ctrl) for 7 days. Quantitative analysis of GLUT4 relative to α-tubulin obtained from western blot. (**E**) Fatty acid composition in the medium was analyzed using gas chromatography. (**F**) Quantification of *Elovl* family gene expression by RT-qPCR. (**G**) Pathway of glucose assimilation to oleic acid. Data are mean ± SEM. $n$ = 3–6 independent replicates. Each replicate experiment with 3–6 wells containing pooled cells from 3 to 5 mice or medium. (**A**, **B**) The glucose uptake and lipid measurements were normalized based on cell count and viability. A two-way ANOVA was performed. Both the Indinavir and testosterone treatments were significant. The interaction was significant. Tukey's Honest Significant Difference test was performed as a post hoc test. Different letters represent significantly different groups. Student's $t$-test was used for comparison between the two groups. Data were considered statistically significant at $p < 0.05$.

The online version of this article includes the following source data and figure supplement(s) for figure 6:

**Source data 1.** Raw data and sample metadata for *Figure 6*.

**Source data 2.** PDF file containing original western blots for *Figure 6D*, indicating the relevant bands and treatments.

**Source data 3.** Original files for western blot analysis displayed in *Figure 6D*.

**Figure supplement 1.** Oleic acid synthesis in vivo regulated by the testosterone–androgen receptor system.

**Figure supplement 1—source data 1.** Raw data and sample metadata for *Figure 6—figure supplement 1*.

**Figure supplement 2.** The effects of lipid mixture (LM) on sperm function.

**Figure supplement 2—source data 1.** Raw data and sample metadata for *Figure 6—figure supplement 2*.

**Figure supplement 2—source data 2.** Results of the reanalysis of *Figure 6—figure supplement 2* using nonparametric tests and effect sizes with 95% confidence intervals are excerpted from the response to the reviewer.

sperm at any concentration compared with the control (LM 0%). However, VSL was significantly increased at 1% LM.

## ACLY expression is upregulated by testosterone and is essential for the metabolic shift that is associated with increased linear motility

ACLY is upregulated in seminal vesicle epithelial cells as a factor that enables the utilization of citrate from the TCA cycle for fatty acid synthesis outside the mitochondria. Therefore, we performed shRNA knockdown experiments on ACLY to clarify the impact of ACLY on the cellular metabolic shift. The addition of testosterone significantly increased the amount of ACLY protein in seminal vesicle epithelial cells (*Figure 7A, B*). Viral infection with shRNAs encoding *Acly*-targeted shRNAs markedly decreased the amount of ACLY (*Figure 7C*). When ACLY was knocked down using shRNA, the proliferation arrest caused by testosterone was released, and the cells began to proliferate again (*Figure 7D*). This knockdown of ACLY increased the oxygen consumption of seminal vesicle epithelial cells in the control and testosterone-treated groups (*Figure 7E, F*). Thus, ACLY expression is associated with mitochondrial metabolic pathway activity in seminal vesicle epithelial cells. Furthermore, the knockdown of ACLY also significantly suppressed testosterone-induced fatty acid synthesis in seminal vesicle epithelial cells (*Figure 7G*). According to the GC–MS quantification results, the amount of fatty acids secreted into the culture medium was also reduced by ACLY knockdown. In particular, the enhancement of oleic acid secretion by testosterone was inhibited by ACLY knockdown (*Figure 7H–J*). Supernatants from epithelial cells of the testosterone-added group induced motility and linear motility (highest VSL and LIN; *Figure 7K–N*). However, the knockdown of ACLY significantly reduced the effects of the culture supernatant on sperm motility and LIN (*Figure 7K–N*). These results indicate that cell proliferation stops when a metabolic shift mediated by ACLY occurs, and glucose is reallocated to fatty acid synthesis. Fatty acid synthesis is an important function of the seminal vesicles, and it was found that when testosterone is present, fatty acid synthesis enhancement and cell proliferation stop simultaneously. However, it is not known what effects occur when cell proliferation stops.

## This characteristic metabolic mechanism induced by testosterone is also observed in human seminal vesicle epithelial cells

To determine whether these findings in mice could be applied to humans, the commercially available human seminal vesicle epithelial cells (HSVEpiC) were used for the following studies (*Figure 8A*). The

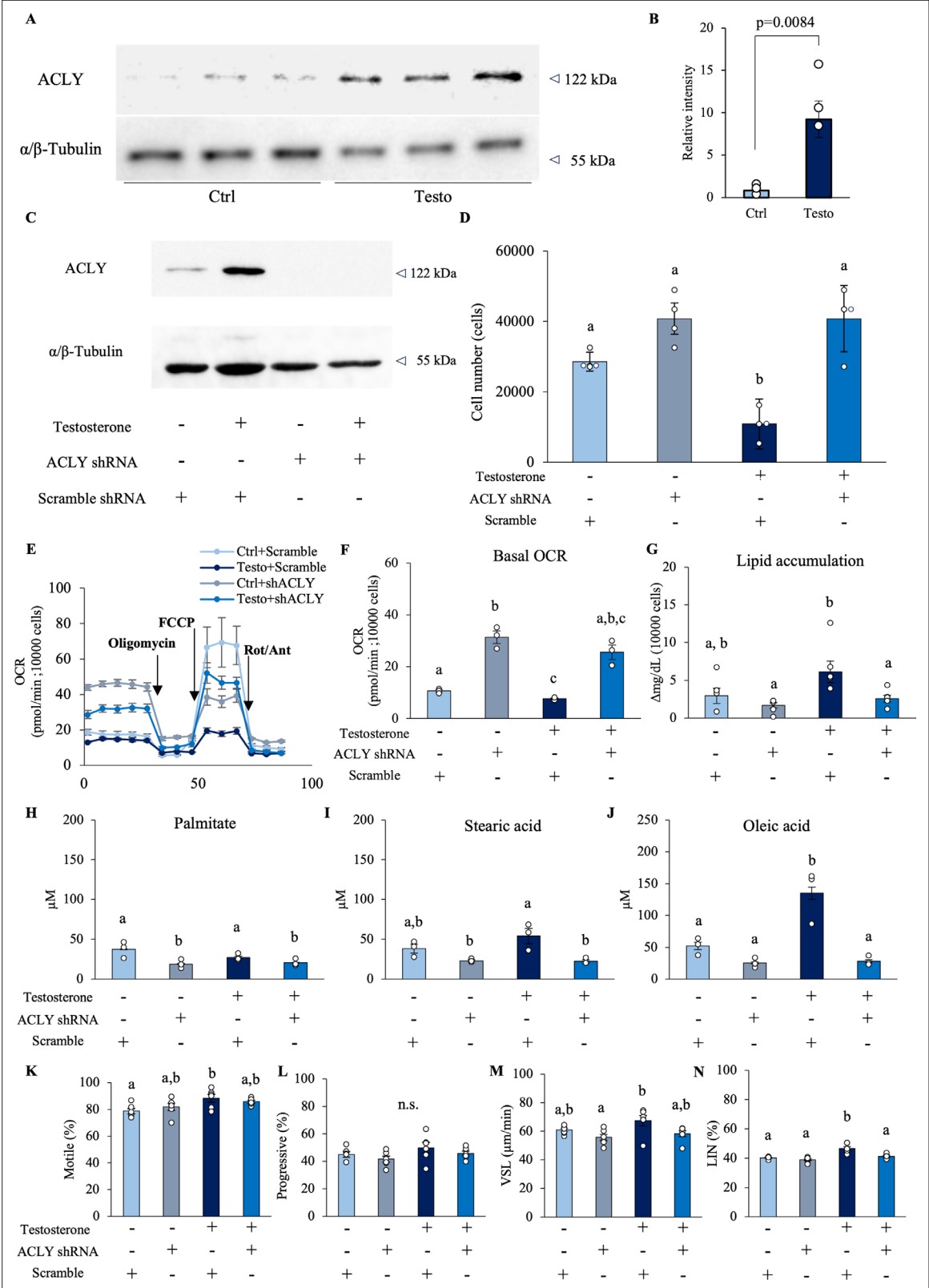

**Figure 7.** Testosterone-regulated ACLY induces metabolic shifts in seminal vesicle epithelial cells. (**A**) Western blot images of ACLY and α/β-tubulin in three sets of seminal vesicle epithelial cells cultured with 100 ng/ml testosterone (Testo) or in vehicle (Ctrl) for 7 days. (**B**) Quantitative analysis of ACLY relative to α-tubulin obtained from western blot. (**C**) shRNA knockdown experiments of ACLY in seminal vesicle epithelial cells. ACLY protein levels in scrambled shRNA or ACLY shRNA-transfected seminal vesicle epithelial cells cultured with or without 100 ng/ml testosterone were determined by

*Figure 7 continued on next page*

*Figure 7 continued*

western blot. (**D**) Number of cells at 7 days after incubation. (**E**, **F**) Changes in oxygen consumption of seminal vesicle epithelial cells were analyzed by a flux analyzer: (**E**) Oxygen consumption rate (OCR) kinetics of seminal vesicle epithelial cells transfected with scrambled shRNA or ACLY shRNA. (**F**) Basal OCR. (**G**) The impact of ACLY knockdown on testosterone-induced fatty acid synthesis was measured. These measurements were normalized based on cell count and viability. The viability of the cells before experiments was 67–81%. (**H–I**) Fatty acid composition in the cultured supernatants was analyzed using gas chromatography. (**H**) Palmitate (C16:0). (**I**) Stearic acid (C18:0). (**J**) Oleic acid (C18:1). (**K–N**) The effect of culture supernatants of testosterone-treated cells on sperm motility was evaluated, especially after ACLY knockdown. (**K**) Motile. (**L**) Progressive motile. (**M**) Straight-line velocity; VSL. (**N**) Linearity; LIN. Data are mean ± SEM. $n$ = 3–6 independent replicates. Repeated experiments were performed with cells recovered from 3 wells containing pooled cells from 3 to 5 mice. Student's $t$-test was used for comparison between the two groups. Percentage data were subjected to arcsine transformation before statistical analysis. (**B**) Significance was tested in comparison using Student's $t$-test. (**D**, **G–M**) A two-way ANOVA was performed. Tukey's Honest Significant Difference test was performed as a post hoc test. (**D**) Both the ACLY knockdown and testosterone treatment were significant. The interaction was significant. (**G**) Both the ACLY knockdown and testosterone treatment were significant. The interaction was not significant. (**H**, **I**) Only the ACLY knockdown was significant. The interaction was not significant. (**J**) Both the ACLY knockdown and testosterone treatment were significant. The interaction was significant. (**F**, **N**) Since the results of the Bartlett test were significant, a two-way ANOVA using a generalized linear model was performed. Games–Howell was performed as a post hoc test. (**E**) Only the ACLY knockdown was significant. The interaction was not significant. (**N**) Testosterone treatment was significant. The interaction was significant. Different letters represent significantly different groups. Data were considered statistically significant at $p < 0.05$.

The online version of this article includes the following source data for figure 7:

**Source data 1.** Raw data and sample metadata for *Figure 7*.

**Source data 2.** PDF file containing original western blots for *Figure 7A*, indicating the relevant bands and treatments.

**Source data 3.** PDF file containing original western blots for *Figure 7C*, indicating the relevant bands and treatments.

**Source data 4.** Original files for western blot analysis displayed in *Figure 7A, C*.

growth curves showed that cell proliferation was inhibited by 100 ng/ml of testosterone compared to the control; a pattern similar to that observed in murine seminal vesicle epithelial cells (*Figure 8B*). The characteristics of metabolic activity of HSVEpiC were also similar to those of mice, with testosterone significantly enhancing the ECAR response (*Figure 8C–E*), but significantly reducing the capacity for oxygen consumption in mitochondria (*Figure 8F–H*). Moreover, fluorescent-glucose uptake was increased by testosterone and decreased by the inhibitor of GLUT4 function (*Figure 8I*). Additionally, while testosterone also significantly increased fatty acid synthesis, this increase was completely abolished by the GLUT4 inhibitor (*Figure 8J*). According to the quantitative results obtained using GC–MS, the amount of fatty acids secreted into the culture medium also decreased due to the inhibition of GLUT4 by indinavir. In particular, it was shown that oleic acid secretion is completely dependent on glucose via GLUT4 (*Figure 8K–M*).

## Discussion

Seminal plasma has a variety of functions, including sperm transport, nutritional support, interactions with the female reproductive tract, and especially the regulation of its relationship with the female immune system. Mammalian seminal plasma is secreted mainly from the seminal vesicle and is rich in cytokines, prostaglandins, and SVS family members (*Gonzales, 2001*). For this reason, recent studies have focused on the immunomodulatory function of seminal plasma in the female reproductive tract. On the other hand, fructose and lipids in seminal plasma are the major energy sources for sperm (*Lenzi et al., 1996*). However, although metabolomic analyses of seminal plasma have shown that metabolic substrates are present at higher concentrations in seminal plasma than in blood (*Rosecrans et al., 1987*; *Mann, 1946*; *Humphrey and Mann, 1949*), how these substrates are synthesized and specifically how they alter sperm motility has not been determined. Moreover, though the motility of ejaculated spermatozoa is known to decrease with aging, few reports have revealed age-related changes in the seminal vesicles, which are responsible for seminal plasma synthesis.

In this study, we focused on the reproductive organs that synthesize seminal plasma as target organs for testosterone and found that factors formed by testosterone-dependent metabolic changes in seminal vesicles activate sperm mitochondria and enhance their linear motility. Testosterone is known to activate intracellular ARs and alter intracellular metabolic pathways in skeletal muscle (*Haren et al., 2011*), prostate cells (*Costello and Franklin, 2002*) and cardiomyocytes (*Wilson et al., 2013*). ARs are known to directly regulate the expression of glycolytic and lipid metabolic genes (*Wilson et al.,*

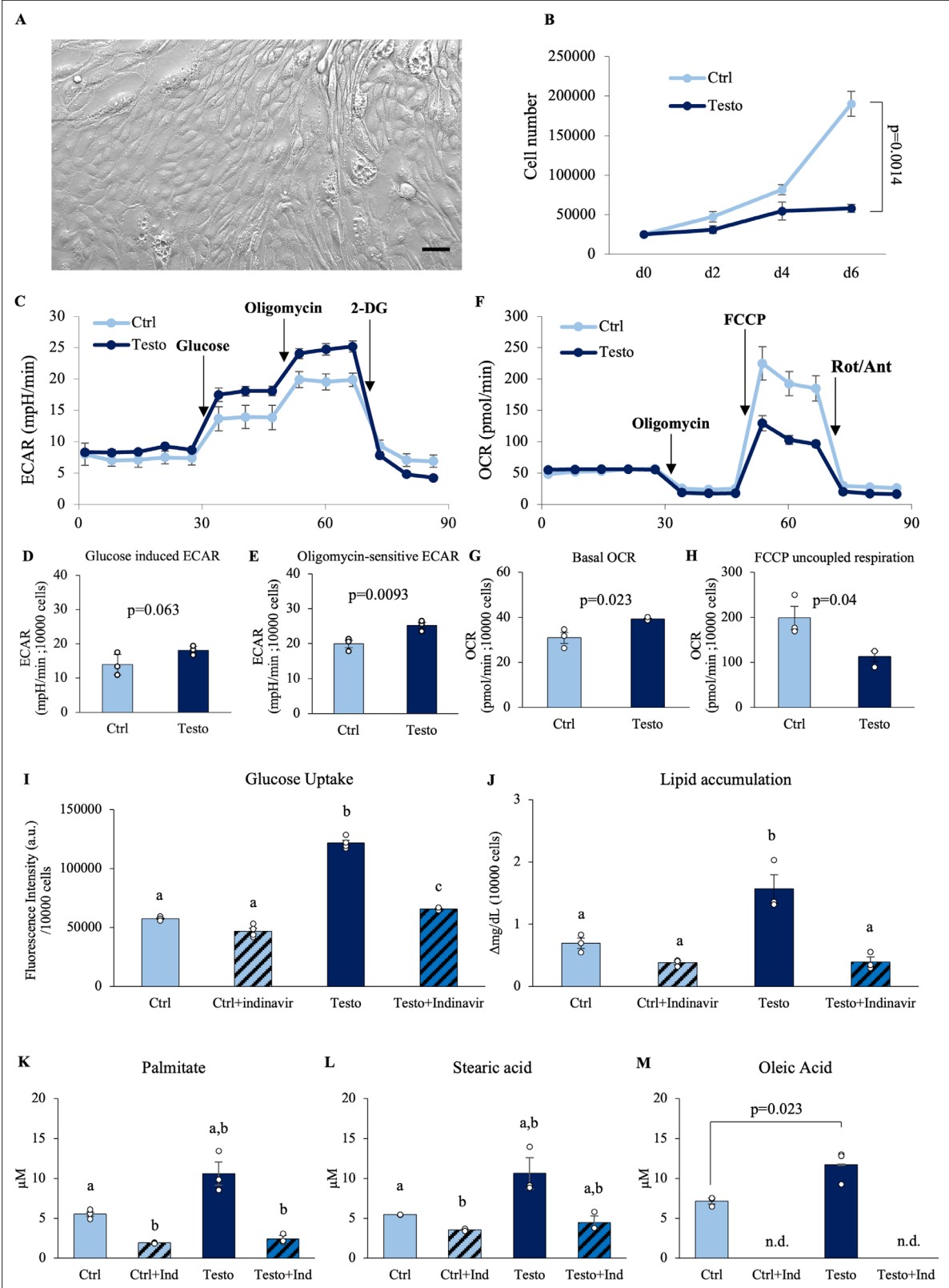

**Figure 8.** Testosterone regulates the metabolic activity of human seminal vesicle epithelial cells. (**A**) Representative image of human seminal vesicle epithelial cells. (**B**) Growth curves of seminal vesicle epithelial cells with 100 ng/ml testosterone (Testo) or without (Ctrl). (**C**–**H**) Extracellular acidification rate (ECAR) kinetics and oxygen consumption rate (OCR) kinetics of human seminal vesicle epithelial cells after 6 days of culture with or without 100 ng/ml testosterone using a flux analyzer. (**D**) Glucose-induced ECAR and (**E**) oligomycin-sensitive ECAR. (**G**) Basal OCR and (**H**) FCCP uncoupled respiration.

*Figure 8 continued on next page*

*Figure 8 continued*

(**I**) The glucose uptake ability of seminal vesicle epithelial cells was detected using fluorescence-tagged glucose. (**J**) Lipid accumulation in the medium where human seminal vesicle epithelial cells were incubated for 24 hr. (**K–M**) Fatty acid composition in the cultured supernatants was analyzed using gas chromatography. (**K**) Palmitate (C16:0). (**L**) Stearic acid (C18:0). (**M**) Oleic acid (C18:1). Data are mean ± SEM. $n$ = 3 independent replicates. Student's $t$-test was used for comparison between the two groups. (**C–J**) These measurements were normalized based on cell count and viability. The viability of the cells before experiments was 88–96%. (**I**, **K**, **L**) Since the results of the Bartlett test were significant, a two-way ANOVA using a generalized linear model was performed. Games–Howell was performed as a post hoc test. Both the Indinavir and testosterone treatment were significant. The interaction was significant. (**J**) A two-way ANOVA was performed. Tukey's Honest Significant Difference test was performed as a post hoc test. Both the indinavir and testosterone treatments were significant. The interaction was significant. Different letters represent significantly different groups. Data were considered statistically significant at $p < 0.05$.

The online version of this article includes the following source data for figure 8:

**Source data 1.** Raw data and sample metadata for *Figure 8*.

*2013*; *Gonthier et al., 2019*; *Troncoso et al., 2021*). In the present study, the nuclear localization of AR in seminal vesicle epithelial cells (i.e., functioning as a transcription factor) and the activation of the glycolytic system by testosterone were similar to those in the above cell types. In seminal vesicle epithelial cells, an increased trend in the expression of *Slc2a3* and an enhancement of GLUT4 function were observed in a testosterone-dependent manner. Such effects of testosterone on glucose metabolism have also been reported in skeletal muscle, liver, and adipose tissue, where testosterone leads to the phosphorylation of GLUT4 and enhances its translocation to the plasma membrane (*Chen et al., 2006*; *Sato et al., 2008*). This is consistent with the results of our metabolic flux analyses, which show that testosterone increases the rate of glycolysis in seminal vesicle epithelial cells by increasing glucose uptake. We further confirmed that GLUT4 is the transporter that mediates this glucose uptake using indinavir, a GLUT4-specific inhibitor.

This study suggests that fatty acid synthesis is an important function of the seminal vesicles, which provide an environment where the ejaculated sperm can carry out active metabolism. The shift to fatty acid synthesis was consistent with the cessation of cell proliferation in the presence of testosterone. This mechanism could prevent the seminal vesicles from excessively consuming the substrates necessary for sperm motility. Manipulation of ACLY expression in mouse myoblasts and embryonic stem cells has been shown to alter the TCA cycle into a 'non-normal cycle' (*Arnold et al., 2022*). One advantage of the non-normal TCA cycle is its ability to retain carbon instead of combusting it and to regenerate the cytosolic NAD+ needed to maintain glycolysis (*Arnold et al., 2022*; *Luengo et al., 2021*). It suggests that ACLY, activated by testosterone, may suppress proliferation by skipping several steps in the mitochondrial TCA cycle and minimizing ATP production associated with aerobic respiration in seminal vesicle epithelial cells.

Testosterone has been reported to cause hypertrophy of cardiac myocytes (*Yeap, 2015*; *Marsh et al., 1998*). Cell hypertrophy is partly due to the accumulation of fatty acids in the cytoplasm, which causes significant damage to cellular functions (*Grabner et al., 2021*; *Mittendorfer, 2011*). On the other hand, normal cell hypertrophy, especially in myofibers, involves the activation of signaling molecules such as Myc, hypoxia-inducible factor, and Pi3k–Akt–mTor, which regulate metabolic reprogramming in cancer by increasing glucose uptake in the presence of oxygen (*Wackerhage et al., 2022*; *DeBerardinis and Chandel, 2016*; *Semenza, 2012*). In the present study, induction of ACLY expression by testosterone not only decreased cycling in the TCA cycle but also promoted glucose assimilation, which played an important role in fatty acid synthesis. Thus, in tissues where fatty acid synthesis is promoted, either β-oxidation-dependent ATP production in mitochondria or the secretion of fatty acids occurs to maintain cellular function (*Morant-Ferrando et al., 2023*; *Duta-Mare et al., 2018*; *Huang et al., 2014*), providing evidence to explain why the presence of fatty acids in the cell has no adverse effect. In accessory reproductive glands, such as the prostate and seminal vesicles, glucose uptake is activated in a testosterone-dependent manner. However, mitochondrial respiration is not enhanced, cell proliferation is not induced, and the metabolites are used as a substrate for sperm motility. This may be because, in the prostate, pyruvate synthesized by the glycolytic system in response to testosterone is converted to acetyl CoA, which is then converted to citrate in the TCA cycle, citrate is released as the main component of seminal plasma (*Costello and Franklin, 2002*).

The testosterone-dependent growth of cancerous prostate epithelial cells may be due to genetic mutations, such as elevated expression of the gene encoding aconitase, which converts citrate to

isocitrate, which blocks the TCA cycle and inhibits citrate secretion (*Tsui et al., 2011*). On the other hand, in the analysis of seminal vesicle epithelial cells, testosterone treatment induced the expression of *Acc*, which encodes the rate-limiting enzyme for fatty acid synthesis, and *Elovl6*, which elongates C16:0 to C18:0, thereby supplying substrate for oleic acid production, significantly increasing total saturated fatty acid and oleic acid content. Therefore, testosterone promotes glucose conversion into fatty acids, especially oleic acid, in seminal vesicle epithelial cells. Although there is a commonality in the activation of the glycolytic system in testosterone target tissues, the different expression patterns of genes involved in metabolism in mitochondria may exert their specific functions in each tissue. Among fatty acid synthesis genes, *Acc*, *Fasn*, and *Acly* are localized to syntenic regions on human chromosome 17q and mouse chromosome 11, respectively, suggesting that this chromosomal locus is a coordinated regulatory hub that promotes expression of lipogenic genes in seminal vesicle epithelial cells. The co-regulation of these genes in seminal vesicle epithelial cells leads to the synthesis of fatty acids.

Decreased testosterone synthesis in the testes is a phenomenon observed with aging associated with decreased spermatogenesis (*Almeida et al., 2017*). However, the probability of obtaining sperm without ICSI or IVF is extremely low, despite the presence of sufficient numbers of sperm with normal morphology (*Murata et al., 2014*; *Stone et al., 2013*; *Frattarelli et al., 2008*). This indicates that testosterone plays a major role not only in spermatogenesis but also in the functional maturation of sperm. In this study, metabolic abnormalities induced by inhibition of androgen receptor function caused abnormal proliferation of seminal vesicle epithelial cells in aged mice. In other words, the findings of the seminal vesicle epithelial cell culture experiments and flutamide treatment may mimic the changes that occur with aging. Oleic acid synthesis by testosterone-dependent metabolic pathways was regulated in seminal vesicle epithelial cells. It could be assumed that a decrease in testosterone levels causes abnormalities in the components of human semen. In particular, the results open up the possibility of a translational study to detect low testosterone levels in men by testing for oleic acid in semen.

Because seminal plasma contains components specific to certain male reproductive organs, differences in protein composition may reflect pathological processes in these specific organs (*Drabovich et al., 2014*). However, although proteomic and metabolomic analyses of seminal plasma have been performed (*Smyth et al., 2022*; *Drabovich et al., 2014*), the functional changes in seminal plasma, especially the effects on in vivo fertilization potential, are not fully understood. This is because there are many challenges to measuring the direct relationship between sperm metabolism and movement patterns. Although our findings and previous studies suggest that oleic acid may play an important role in improving fertility, it is still too early to draw a definitive conclusion from this study. If a direct causal relationship between fatty acid composition and improved sperm motility is established, differences in the concentration of fatty acids in the seminal plasma will become a predictive marker of the fertilizing ability of semen. The interspecies differences between humans and mice should be interpreted with caution, and the translational aspects to humans are speculative, but this study revealed a testosterone-dependent mechanism of oleic acid production in seminal vesicle epithelium.

Overall, the results of the studies described herein strongly support the hypothesis that testosterone-dependent glucose to oleic acid conversion processes are required for normal seminal plasma functions in the seminal vesicle (*Figure 9*). This study advances our understanding of the role of testosterone in glucose metabolism and fatty acid synthesis and lays the groundwork for future research in reproductive biology and fertility.

## Materials and methods
### Study design
The objective of this study was to elucidate the mechanisms of seminal plasma synthesis. Testosterone-dependent changes in the seminal plasma function were examined by comparative analysis of adult mice and flutamide-treated mice. The bioassays of extracts from male accessory glands by epididymal sperm motility analysis were used to examine the seminal plasma components important for sperm function. We also isolated the mouse seminal vesicle epithelial cells and cultured them and investigated the synthesis mechanism of the seminal plasma components using RNA-seq, FluxAnalyzer, and shRNA knockdown. Finally, we performed validation using the HSVEpiC.

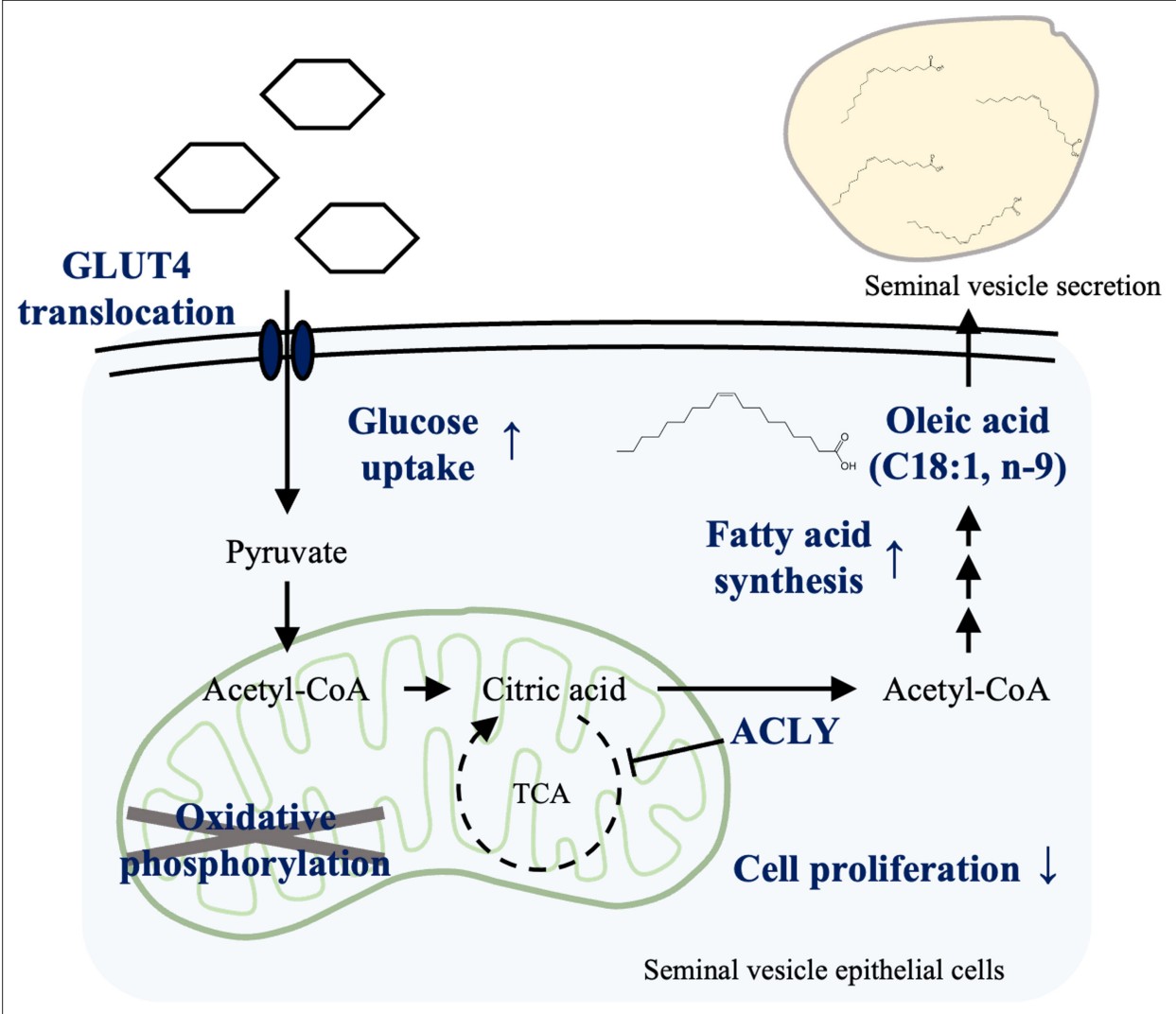

**Figure 9.** Testosterone-induced metabolic reprogramming in seminal vesicle epithelial cells. Testosterone promotes glucose uptake by changes in the localization of GLUT4 in seminal vesicle epithelial cells. ATP-citrate-lyase (ACLY) activated by testosterone promotes fatty acid synthesis by skipping some tricarboxylic acid (TCA) cycle steps. Simultaneously, cell proliferation in the epididymal epithelial cells is suppressed. Testosterone also promotes the synthesis of oleic acid from acetyl-CoA synthesized by ACLY.

## Chemicals and animals

Unless otherwise noted, chemicals used in this study were purchased from Sigma-Aldrich (St. Louis, MO, USA) or Nacalai Tesque (Osaka, Japan). Equine chorionic gonadotropin (eCG) and human chorionic gonadotropin (hCG) were purchased from Asuka Seiyaku (Tokyo, Japan). All procedures during animal experiments were reviewed and approved by the Animal Care and Use Committee of Hiroshima University (Hiroshima, Japan; C21-9-3) and conducted according to regulations. Specific pathogen-free C57BL/6NCrl male mice or Crl: CD1 (ICR) female mice were purchased from Jackson Laboratory Japan (Kanagawa, Japan) and housed in an environmentally controlled room with a 12-hr light/dark cycle, a temperature of 23 ± 3°C, and free access to laboratory food (MF; Oriental Yeast Co, Ltd, Tokyo, Japan) and tap water. Eight- to sixteen-week-old male or female mice were used for the experiments. Flutamide-treated mice were injected subcutaneously for 7 days with 50 µg/g (body weight) of flutamide dissolved in corn oil. Vehicle-treated mice received 100 µl of corn oil subcutaneously for 7 days. The aged males were 12–16 months old.

## Bioassay using pseudo-seminal plasma and epididymal sperm

Prostate (mixed anterior, dorsal, lateral, and ventral) and seminal vesicles were collected from adult C57BL/6NCrl mice. Seminal vesicle secretions were obtained by pressing the seminal vesicles immediately after euthanasia. Prostatic extracts were obtained by homogenizing the prostate immediately after euthanasia. Seminal vesicle secretions and prostate extracts collected from one male mouse were dissolved in 500 µl and 200 µl of mHTF medium without BSA (*Umehara et al., 2020*), respectively, then centrifuged at 2000 × *g* for 5 min at room temperature, and the supernatant was collected, and protein levels were determined. The seminal vesicle secretions from flutamide-treated mice were also collected similarly.

Cauda epididymis sperm recovered from male mice was dispersed in 500 µl of mHTF without BSA and centrifuged at 300 × *g*, 3 min to wash out epididymis-derived factors. The final protein concentration of prostate and seminal vesicle secretion was adjusted to 10 mg/ml. An aliquot (2 ml) of the pooled prostate or seminal vesicle extracts was used to measure viscosity with a tuning-fork vibro viscometer (SV-1A, A&D Company, Tokyo, Japan). The epididymal sperm were mixed with seminal vesicle secretion and/or prostate extract as shown in *Figure 1*. The samples were incubated at 37°C for 60 min in a humidified atmosphere of 5% $CO_2$ in air. Sperm motility patterns were then examined using computer-assisted sperm analysis (CASA; HT CASA-Ceros II, Hamilton Thorne, Beverly, MA, USA), with software version 1.11.9. The following sperm motility parameters were analyzed: percentage of motile sperm (denoted by an average path velocity >40 µm/s and a VSL >20 µm/s), percentage of motile sperm with progressive motility (denoted by an average path velocity >40 µm/s and a straightness ratio >60%). Straight-line velocity; VSL: The distance between the first and last tracking points of the sperm trajectory divided by the elapsed time. Curvi-linear velocity; VCL: The sum of the distances between the centers of brightness in each frame divided by the elapsed time. Linearity: LIN: Measures the straightness of the path. LIN is derived from the ratio of VSL/VCL multiplied by 100.

## Serum testosterone levels

The serum was separated from the clot by centrifugation at 1500 × *g* for 15 min. Testosterone levels in the serum samples were determined by a rodent testosterone ELISA test kit (ERKR7016, Endocrine Technologies, Inc, Newark, CA, USA) according to the manufacturer's manual.

## Mitochondrial activity

Mitochondrial activity of sperm was measured using a MitoPT JC-1 Assay Kit (911, Immuno Chemistry Technologies, LLC, Bloomington, MN, USA) according to our previous study (*Zhu et al., 2019*). Briefly, sperm were incubated with 200 µl of working solution containing 5,5',6,6'-tetrachloro-1,1',3,3'-tetraethylbenzimidazolyl carbocyanine iodide (JC-1) dye at 37°C for 30 min in the dark. Using 10 µM of antimycin control (A8674; Sigma) under the same incubation conditions, we confirmed that the JC-1 reaction is sensitive to changes in membrane potential. The sperm suspension was centrifuged and washed twice with the base medium. After washing, the sperm pellet was resuspended in the base medium and analyzed with an Attune NxT Acoustic Focusing Cytometer (Thermo Fisher Scientific Inc, Waltham, MA, USA) using a 488-nm laser and a filter with a bandwidth of 574/26 nm. The intensity of the average value was analyzed as the mean fluorescence intensity of JC-1 orange aggregates. A total of 100,000 sperm events were analyzed. The sperm population is shown in *Figure 1—figure supplement 3*.

## In vitro fertilization

Three-week-old CD1 female mice were injected intraperitoneally with 4 IU eCG to stimulate follicle development and with 5 IU hCG 48 hr later. Cumulus oocyte complex (COCs) collected from the oviductal ampulla 16 hr after hCG injection were placed into 100 µl of HTF medium. Sperm were collected from the cauda epididymis of 15 weeks or older C57BL/6N male mice in 100 µl of HTF medium. After 60 min of incubation with or without treatments (Ctrl, Flu, or HTF), the sperm were added into HTF medium at a final concentration of $2 \times 10^5$ spermatozoa/mL, and coincubated with the COCs. After 6 hr incubation, oocytes were washed three times thoroughly and cultured in 250 µl of KSOM medium (MR 106-D; Sigma). The cleavage rate was determined on day 1 (day 0 = day of IVF), and the appearance of blastocysts was recorded on day 5.

## Immunohistochemistry and immunofluorescence

Seminal vesicles were fixed overnight in 4% (wt/vol) paraformaldehyde/phosphate-buffered saline (PBS), dehydrated in 70% (vol/vol) ethanol, and embedded in paraffin. Paraffin-embedded fixed sections (4 µm) were deparaffinized and hydrated with xylene and ethanol, respectively. Some slides were stained with hematoxylin and eosin. For antigen retrieval, sections were placed in citrate buffer (pH 6.0), microwaved until boiling, and allowed to stand at the temperature of pre-boil (95°C) for 15 min. Sections were incubated with 3% (vol/vol) hydrogen peroxide in methanol to block endogenous peroxidase and then incubated with Normal Goat Serum Blocking Solution for rabbit antibody (S-1000, Vector Laboratories, Newark, CA, USA) to block nonspecific sites. The sections were then incubated overnight with primary rabbit antibodies: anti-AR antibody (1:100; ab133273; Abcam, Cambridge, UK), anti-Ki67 antibody (1:1000; ab15580; Abcam), or anti-ACLY antibody (1:200; ab40793; Abcam). Positive signals were visualized by DAB (3,3′-Diaminobenzidine, 040-27001, Fujifilm Wako, Osaka, Japan) using the VECTASTAINE LifeABC kit (PK-6101, Vector Laboratories) according to the manufacturer's protocol. Apoptotic cells (including fragmented DNA) were detected using the In Situ Cell Death Detection Kit POD (11684817910, Roche Ltd, Basel, Switzerland). To avoid false-positive or false-negative results, DNase-treated sections were used as positive controls, and TdT-untreated sections were used as negative controls for apoptosis.

Nuclei were visualized with hematoxylin (Sakura Finetek, Tokyo, Japan). Ki67 and apoptotic positive cells were quantified using ImageJ software (National Institutes of Health, Bethesda, MD, USA), with software version 2.1.0/1.53u. Randomly selected sections were stained, and images were obtained from randomly selected fields in each section. The positive percentage is defined as the percentage of positively stained cells in the total number of epithelial cells evaluated. Only positive staining was considered, regardless of staining intensity. For immunofluorescence, after blocking nonspecific sites, the sections were incubated with a primary antibody: anti-GLUT4 antibody (1:200; ab33780; Abcam). After washing with 0.3% (vol/vol) Triton X-100 in PBS (-), Cy3-labeled goat anti-rabbit IgG (1:200, C-2306, Sigma) and DAPI (VECTESHIELD Mounting Medium with DAPI, H-1200, Vector Laboratories) were used to visualize the antigen and nuclei. Digital images were taken using an APX100 Digital Imaging System and CellSens imaging software (EVIDENT, Tokyo, Japan), with software version 4.1.1.

## Isolation and culture of seminal vesicle epithelial cells

Seminal vesicles were removed from euthanized mice, cut open vertically to expose the seminal vesicle cavity, and incubated with Hanks' balanced salt solution (HBSS; $CaCl_2 \cdot 2H_2O$ 0.185 g/l, $MgSO_4$ 0.098 g/l, KCl 0.400 g/l, $KH_2PO_4$ 0.060 g/l, $NaHCO_3$ 0.350 g/l, NaCl 8.000 g/l, $Na_2HPO_4$ 0.048 g/l, D-glucose 1.0 g/l, dissolved in ultrapure water). Tissues were digested with 10 ml of HBSS containing 25 mg/ml pancreatin and 0.25% (vol/vol) trypsin EDTA for 1 hr at 4°C, 45 min at room temperature, and 15 min at 37°C, and epithelial cells were obtained. These were passed through a 70-µm nylon filter (352350, Corning Inc, Corning, NY, USA) and seeded into collagen-coated 12-well plates (4815-010, IWAKI, Shizuoka, Japan). Cells were cultured in DMEM/F-12, 10% fetal bovine serum (FBS; 10438-018, Thermo Fisher Scientific Inc), 100 U/ml penicillin, 100 µg/ml streptomycin. The culture conditions were 37°C, 5% $CO_2$, 95% air, and humidity.

HSVEpiC (4460, ScienCell Research Laboratories, Carlsbad, CA, USA) were purchased, seeded in 2 µg/cm² poly-L-lysine-coated plates, and cultured in epithelial cell medium (4101, ScienCell Research Laboratories). The cells were seeded at 10,000 cells/cm² on collagen-coated plates the day before 100 ng/ml testosterone treatment. Testosterone was dissolved in ethanol, and the final concentration of ethanol was adjusted to 0.1% (vol/vol) when used for cultured cells. The culture medium was replaced every 48 hr.

According to the manufacturer's product sheet, HSVEpiC are cryopreserved at passage 1, characterized by immunofluorescence using antibodies against cytokeratin-18 and/or cytokeratin-19, and are negative for HIV-1, HBV, HCV, mycoplasma, bacteria, yeast, and fungi. Consistent with this information, we confirmed epithelial morphology and cytokeratin immunoreactivity in our cultures.

Because HSVEpiC are primary human cells for which standard short tandem repeat (STR) profiling panels are not available, we did not perform additional STR-based authentication. Instead, cell identity was verified by the documented tissue of origin, vendor characterization, and reproducible expression of epithelial markers.

Primary mouse seminal vesicle epithelial cells were freshly isolated from C57BL/6NCrl males as described above and were not propagated beyond passage 2. Cell identity was verified by anatomical origin, characteristic epithelial morphology, and cytokeratin immunostaining (>80% positive). A stromal cell marker (vimentin) was also used to confirm purity, but only a few of the cells were positive. The contaminating cell type was considered to be mainly muscle cells because the gene expression levels of muscle cell markers verified by RNA-seq were relatively high.

All cell cultures used in this study were routinely tested for mycoplasma contamination by MycoAlert Mycoplasma Detection Kit (Cat. #LT07-218, Lonza, Basel, Switzerland). The most recent test before the experiments reported here (November 2025) was negative.

None of the cell lines used in this study are listed in the International Cell Line Authentication Committee Register of Misidentified Cell Lines (version 13, April 2024).

## Cell proliferation curve

The seminal vesicle epithelial cells were seeded at 10,000 cells/cm$^2$ and then were cultured in various concentrations of testosterone. The cells were washed with HBSS (without CaCl$_2$–2H$_2$O/MgSO$_4$) at the time of cell counting and collected using 0.25% (vol/vol) trypsin. The recovered cells were suspended in DMEM/F-12 medium, and cell numbers were determined using an Automated Cell Counter TC20TM (Bio-Rad Laboratories, Hercules, CA, USA) to calculate the number of cells contained per well (4 cm$^2$).

## Cell cycle analysis

The recovered cells as described above were fixed in 2–5 ml of cold (4°C) 70% methanol for at least 30 min on ice. After centrifugation at 500 × $g$ for 5 min at 4°C to obtain pellets, the cell pellet was suspended in 50 µM propidium iodide and 20× diluted RNase (19101, QIAGEN Inc, Venlo, Netherlands) and then was incubated at 37°C for 30 min. After twice washing with 1 ml of PBS at 300 × $g$ for 3 min at 4°C, the cells were analyzed by flow cytometry. The cell population is shown in *Figure 2— figure supplement 1*. A total of 20,000 events were analyzed.

## RNA sequencing

Total RNA was extracted from cells using the RNeasy Mini Kit (74106, QIAGEN). Three independent samples were assayed to evaluate the reproducibility of the experimental procedures. The quality of total RNA (1 µg) submitted for sequencing was checked using a Bioanalyzer (Agilent Technologies Inc, Santa Clara, CA, USA), and all RNA integrity numbers were >7.0. The libraries for RNA sequencing were sequenced with 2 × 150 bp paired-end reads on a NovaSeq6000 (Illumina Inc, San Diego, CA, USA). The obtained sequencing data were analyzed using RaNA-seq (*Prieto and Barrios, 2019*), an open bioinformatics tool for rapid RNA-seq data analysis. FASTQ files were preprocessed in the pipeline using the Fastp tool, and expression was quantified using Salmon. DESeq2 was used to identify differentially expressed genes (adjusted p < 0.05) between control and testosterone-treated cells. Gene expression levels are reported as transcripts per million (TPM). The pathways were identified from the KEGG database on RaNA-seq.

## Extracellular flux analysis

Oxygen consumption rate (OCR) and ECAR were measured using an extracellular flux analyzer (XF HS Mini; Agilent Technologies). Eight-well plates for the XF HS Mini were coated with fibronectin (5 µg/ml, F1141, Sigma-Aldrich) dissolved in PBS, incubated overnight, and then washed with Roswell Park Memorial Laboratory medium for XF (Roswell Park Memorial Institute medium; RPMI, 103576-100, Agilent Technologies; containing 1% FBS) for XF before analysis. Mouse seminal vesicle epithelial cells or HSVEpiC treated with 100 ng/ml testosterone for 7 days were suspended in RPMI medium at a concentration of 10,000 cells/180 µl, or mouse seminal vesicle epithelial cells infected with shRNA were suspended at a concentration of 2000 cells/180 µl in each well. After centrifugation at 300 × $g$ for 3 min, the plates were placed in an incubator at 37°C and 100% air and used for analysis within 1 hr.

For glucose metabolism assessment, cells were sequentially treated as indicated with D-glucose (10 mM; 103577-100, Agilent Technologies), Oligomycin (1 µM; O4876, Sigma-Aldrich), and 2-deoxy-glucose (50 mM; D8375, Sigma-Aldrich).

For mitochondrial respiration measurement, cells were sequentially treated as indicated with Oligomycin (1 µM), carbonyl cyanide 4-(trifluoromethoxy) phenylhydrazone (FCCP; 5 µM, SML2959, Sigma-Aldrich), and Rotenone (1 µM; R8875, Sigma-Aldrich)/Antimycin A (1 µM; A8674, Sigma-Aldrich). Cellular metabolism readouts, such as glycolysis, glycolytic reserve, basal OCR, maximum respiration, ATP production, spare respiratory capacity, were determined using the Agilent Seahorse Wave software with software version 3.0.0.41. However, to avoid many ambiguities and problems with interpretation, 'glycolysis' is written as 'Glucose-induced ECAR', 'glycolytic reserve' as 'Oligomycin-sensitive ECAR', 'maximum respiration' as 'FCCP uncoupled respiration', and 'ATP production' as 'Oligomycin-sensitive respiration'.

## Measurement of pyruvate and lactic acid

For measurements, mouse or HSVEpiC were treated with 100 ng/ml testosterone for 6 days. The culture supernatant was collected after 24 hr of additional culture. Pyruvate consumption was measured using the Pyruvate Assay kit (MET-5029, Cell Biolabs, Inc, San Diego, CA, USA). The fluorescence intensity was measured using a microplate reader (ex. 570 nm/em. 615 nm). Lactate released into the medium was measured using the Lactate Assay kit (L256, DOJINDO LABORATORIES Co, Ltd, Kumamoto, Japan). The absorbance was measured using a microplate reader (wavelength = 450 nm).

## qPCR

The cDNA was synthesized from 40 ng of total RNA using oligo(dT)$_{15}$ primers (3805, Takara Bio Inc, Shiga, Japan) and an Avian myeloblastosis virus (AMV) reverse transcriptase from Promega (M5101, Madison, WI, USA). The cDNA and primers were added to Power SYBR Green PCR Master Mix (4367659, Applied Biosystems, Foster City, CA, USA) in a total reaction volume of 15 µl. Conditions were set to the following parameters: 10 min at 95°C, 15 s at 95°C, and 1 min at 60°C. Cycles were repeated 45 times. Specific primer pairs were selected and analyzed, as shown in *Supplementary file 1*. Expression was first normalized to housekeeping gene *Rpl19*, and fold change was calculated relative to the mean of the control samples.

## Glucose uptake test of seminal vesicle epithelial cells and lipid determination of culture supernatants

Mouse or HSVEpiC treated with or without 100 ng/ml testosterone for 6 days were used for this experiment. Cells were treated with or without the GLUT4 inhibitor indinavir (100 nM) and 100 ng/ml testosterone for an additional 24 hr. Collected 15,000 cells were subjected to Glucose Uptake Assay Kit-Green (UP02; DOJINDO), and fluorescence intensity was measured using a microplate reader (ex. 485 nm/em. 535 nm).

Total lipid content in cultured medium was quantified using a lipid quantification kit (STA-617 Fluorometric, Cell Biolabs, Inc) according to the manufacturer's protocol. The fluorescence intensity was measured using a microplate reader (ex. 485 nm/em. 535 nm).

## GC–MS analysis

Fatty acid measurements in culture supernatant and seminal vesicle secretions were done according to our previous work (*Islam et al., 2021*). Fatty acids in the samples were extracted and methylated using a fatty acid methylation kit (06482-04, Nacalai Tesque) and then purified using a fatty acid methyl ester purification kit (06483-94, Nacalai Tesque). The methylated sample (3.0 ml) was dried and dissolved in 50 µl of elution solution. Samples were subjected to GC–MS.

Fatty acids were determined on an Agilent 7890A GC system coupled with a JMS-T 100 GCv mass detector at the Natural Science Research Center, Hiroshima University (N-BARD). Sample (1.0 µl aliquot) was injected to SP-2560 capillary GC column (100 m × 0.25 mm i.d. × 0.2 µm film thickness; 24056, Sigma-Aldrich). For measurements, helium was used as the carrier gas at a constant flow rate of 2.0 ml/min. The GC oven temperature was gradually increased from 100 to 300°C at a rate of 10°C/min and held at 300°C for 20 min. Ionization was performed using electron ionization with an electron energy of 70 eV. Mass spectra were obtained in scan mode (mass scan range 29–800 *m/z*). NIST MS Search v.2.0 was used to detect and identify fatty acids.

## Sperm motility parameters with the LM

Sperm recovered from the cauda epididymis of male mice were dispersed in 500 µl of mHTF without BSA and centrifuged at 300 × *g* for 3 min to wash out epididymis-derived factors. The sperm pellets

were resuspended in HTF medium with or without an LM that included 2 µg/ml arachidonic acid, 10 µg/ml linoleic acid, 10 µg/ml linolenic acid, 10 µg/ml myristic acid, 10 µg/ml oleic acid, 10 µg/ml palmitic acid, and 10 µg/ml stearic acid (L0288, Sigma-Aldrich). The sperm were then incubated for 60 min at 37°C under a humidified atmosphere of 5% $CO_2$. Sperm motility patterns were subsequently examined using CASA.

### ACLY knockdown experiment using shRNA

The U6 shRNA lentiviral vector of the *Acly* gene (mAcly[shRNA#1]; a virus packaging service, Vector-Builder, Chicago, IL, USA) was used for knockdown experiments. The ACLY sequence targeted by the shRNA vector used in this study was 5′-GCAGCAAAGATGTTCAGTAAA-3′. Lentivirally transduced cells were prepared according to the manufacturer's protocol. Immediately after isolation, the mouse seminal vesicle epithelial cells were cultured overnight in a medium containing 5 µg/ml polybrene (VectorBuilder) and lentiviral particles expressing an shRNA targeting ACLY or a scramble (multiplicity of infection of 10). After an additional overnight incubation in a fresh culture medium, the cells were incubated for 5 days in a medium containing 1 µg/ml puromycin (19752-64, Nacalai Tesque). 100 ng/ml of testosterone was added to all procedures for the testosterone-treated group. Infection efficiency, calculated based on the number of GFP-positive cells using an APX100 Digital Imaging System, ranged between 70% and 90%.

For sperm motility analysis, sperm were collected and washed with mHTF from the culture supernatant and cauda epididymis. The samples were incubated at 37°C for 60 min in a humidified atmosphere of 5% $CO_2$ in air. The sperm motility pattern was then examined using CASA.

### Western blot analysis

Protein samples of mouse seminal vesicle epithelial cells, seminal vesicle tissue, and sperm were prepared by homogenization in cell lysis buffer (04719964001, cOmplete Lysis-M EDTA-free, Roche) and diluted with an equal volume of sample buffer solution containing 2-ME (2×) for SDS–PAGE (30566-22, Nacalai Tesque). The extracts (10 µg protein) were separated by SDS–polyacrylamide gel (10%) electrophoresis and transferred to polyvinylidene fluoride membranes (10600069, Cytiva, Tokyo, Japan). The membranes were then treated with Tris-buffered saline and Tween 20 (TBST, 20 mM Tris, pH 7.5, 150 mM NaCl, 0.1% Tween 20) containing 5% (wt/vol) Non-Fat Dry Milk (MORINAGA MILK INDUSTRY Co, Ltd, Tokyo, Japan) to block nonspecific reactions. The membranes were incubated overnight at 4°C with primary antibodies: anti-GLUT4 antibody (1:100; ab33780; Abcam), anti-ACLY antibody (1:10,000; ab40793; Abcam), or anti-α/β-Tubulin antibody (1:1000; 2148S; Cell Signaling). The next day, membranes were washed with TBST and incubated with HRP-labeled antibody specific for rabbit IgG (1:4000; 7074S; Cell Signaling). The bands were visualized by using Enhanced Chemiluminescence detection systems (RPN2232, Cytiva) and ChemiDoc MP Imaging System (Bio-Rad Laboratories). The bands were analyzed by using the software ImageJ. Densitometry analysis was performed using the ImageJ Gel Analysis tool, and the background of the gel was also removed individually for each band. The full-length blotting images are shown in *Figure 6* and *Figure 7—source data 2 or 3*.

### Statistical analysis

Quantitative data were presented as means ± SEM. The Bartlett's test was used to confirm whether the variances were equal for each group in experiments consisting of three or more groups. All statistical details of the experiments are given in the figure legends. R [version 4.3.1 (2023-06-16)] was used for statistical analysis. Data were considered statistically significant at $p < 0.05$.

## Acknowledgements

The authors would like to thank JoAnne S Richards for carefully proofreading the manuscript and for useful comments. This work was supported in part by JSPS KAKENHI Grant Number 23K14069 to TU and 23H00361 to MS.

# Additional information

## Competing interests

Takashi Umehara: TU holds stocks in and receives a salary from Lullabio Inc as a director and has received honoraria from Rohto Pharmaceutical Co., Ltd. Masayuki Shimada: MS holds stocks in and receives a salary from Hiroshima Cryopreservation Service Co. as a director. In addition, MS has received royalties and grants from Hiroshima Cryopreservation Service Co. M.S. has received consulting fees from Rohto Pharmaceutical Co., Ltd. The other authors declare that no competing interests exist.

## Funding

| Funder | Grant reference number | Author |
| --- | --- | --- |
| Japan Society for the Promotion of Science | JP23K14069 | Takashi Umehara |
| Japan Society for the Promotion of Science | JP23H00361 | Masayuki Shimada |

The funders had no role in study design, data collection, and interpretation, or the decision to submit the work for publication.

## Author contributions

Takahiro Yamanaka, Conceptualization, Data curation, Formal analysis, Investigation, Visualization, Methodology, Writing – original draft, Writing – review and editing; Zimo Xiao, Mahmoud Awad, Investigation, Methodology; Natsumi Tsujita, Formal analysis, Investigation, Methodology; Takashi Umehara, Conceptualization, Funding acquisition, Investigation, Methodology, Writing – review and editing; Masayuki Shimada, Conceptualization, Funding acquisition, Visualization, Project administration, Writing – review and editing

## Author ORCIDs

Takahiro Yamanaka (ID) https://orcid.org/0000-0001-9234-4289
Masayuki Shimada (ID) https://orcid.org/0000-0002-6500-2088

## Ethics

Primary human cells (HSVEpiC; 4460, ScienCell Research Laboratories, Carlsbad, CA, USA) were purchased from a commercial vendor. In accordance with the supplier's ethical policy, donor tissues were collected with written informed consent and have been anonymized. Since researchers did not collect new human data or specimens, no additional ethical review was required based on Hiroshima University's policy. These cells are intended for research use only. Although the vendor's testing indicated negativity for HIV, HBV, and HCV, all procedures followed standard biosafety practices for handling materials of human origin.

All procedures during animal experiments were reviewed and approved by the Animal Care and Use Committee of Hiroshima University (Hiroshima, Japan; C21-9-3) and conducted according to regulations.

Reviewer #1 (Public review): https://doi.org/10.7554/eLife.95541.5.sa1
Reviewer #2 (Public review): https://doi.org/10.7554/eLife.95541.5.sa2
Author response https://doi.org/10.7554/eLife.95541.5.sa3

---

# Additional files

## Supplementary files

Supplementary file 1. Primer sequences used for quantitative real-time PCR.

MDAR checklist

## Data availability

Nucleotide sequence data reported are available in the DDBJ Sequenced Read Archive under the accession number DRA017090. All data generated or analyzed during this study are included in the manuscript and supporting files; source data files have been provided for all figures.

The following dataset was generated:

| Author(s) | Year | Dataset title | Dataset URL | Database and Identifier |
|---|---|---|---|---|
| Yamanaka T, Xiao Z, Tsujita N, Awad M, Umehara T, Shimada M | 2025 | The effects of testosterone on epithelial cells isolated from mouse seminal vesicles | https://ddbj.nig.ac.jp/search/entry/sra-submission/DRA017090 | DDBJ, DRA017090 |

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

# Appendix 1

## Appendix 1—key resources table

| Reagent type (species) or resource | Designation | Source or reference | Identifiers | Additional information |
|---|---|---|---|---|
| Strain, strain background (Mouse, inbred, male) | C57BL/6NCrl | Jackson Laboratory Japan (originated from Charles River) | Strain #:027; RRID:IMSR_CRL:027 | |
| Strain, strain background (Mouse, outbred, female) | Crl: CD1 (ICR) | Jackson Laboratory Japan (originated from Charles River) | Strain #:022; RRID:IMSR_CRL:022 | |
| Cell line (*Homo sapiens*) | Human seminal vesicle epithelial cells | ScienCell | 4460 | |
| Cell line (*Mus musculus*) | Mouse seminal vesicle epithelial cells | This paper | | Isolation and culture conditions are described in the Methods section. |
| Transfected construct (*M. musculus*) | U6 shRNA lentiviral vector targeting Acly | VectorBuilder | mAcly[shRNA#1], ID;VB900215-4814zyj | Target sequence; GCAGCAAAGATGTTCAGTAAA |
| Transfected construct (*M. musculus*) | Scramble control shRNA lentiviral vector | VectorBuilder | Scramble shRNA control, ID;VB010000-9526zpu | Target sequence; CCTAAGGTTAAGTCGCCCTCG |
| Antibody | Rabbit monoclonal anti-AR antibody | Abcam | ab133273, RRID:AB_11156085 | 1:100 |
| Antibody | Rabbit polyclonal anti-Ki67 antibody | Abcam | ab15580, RRID:AB_443209 | 1:1000 |
| Antibody | Rabbit monoclonal anti-ACLY antibody | Abcam | ab40793, RRID:AB_722533 | IF; 1:200, WB; 1:10,000 |
| Antibody | Rabbit polyclonal anti-GLUT4 antibody | Abcam | ab33780, RRID:AB_2191441 | IF; 1:200, WB; 1:100 |
| Antibody | Cy3-labeled goat polyclonal anti-rabbit IgG | Sigma | C-2306, RRID:AB_258792 | 1:200 |
| Antibody | Rabbit polyclonal anti-α/β-Tubulin antibody | Cell Signaling | 2148S, RRID:AB_2288042 | 1:1000 |
| Antibody | Goat polyclonal HRP-conjugated anti-rabbit IgG secondary antibody | Cell Signaling | 7074S, RRID:AB_2099233 | 1:4000 |
| Commercial assay or kit | Rodent testosterone ELISA kit | Endocrine Technologies | ERKR7016 | |
| Commercial assay or kit | MitoPT JC-1 Assay Kit | ImmunoChemistry Technologies | 911 | |
| Commercial assay or kit | In Situ Cell Death Detection Kit, POD | Roche | 11684817910 | |
| Commercial assay or kit | VECTASTAIN Elite ABC kit / LifeABC kit | Vector | PK-6101 | |
| Commercial assay or kit | VECTASHIELD Mounting Medium with DAPI | Vector | H-1200 | |
| Commercial assay or kit | Glucose Uptake Assay Kit-Green | DOJINDO | UP02 | |

*Appendix 1 Continued on next page*

*Appendix 1 Continued*

| Reagent type (species) or resource | Designation | Source or reference | Identifiers | Additional information |
|---|---|---|---|---|
| Commercial assay or kit | Pyruvate Assay Kit | Cell Biolabs | MET-5029 | |
| Commercial assay or kit | Lactate Assay Kit | DOJINDO | L256 | |
| Commercial assay or kit | Total lipid quantification kit | Cell Biolabs | STA-617 | |
| Commercial assay or kit | Fatty acid methylation kit | Nacalai Tesque | 06482-04 | |
| Commercial assay or kit | Fatty acid methyl ester purification kit | Nacalai Tesque | 06483-94 | |
| Chemical compound, drug | Testosterone | Sigma | T-1500 | 100 ng/ml in ethanol |
| Chemical compound, drug | Flutamide | Sigma | F-9397 | Androgen receptor antagonist; 50 µg/g BW, s.c. |
| Chemical compound, drug | eCG | Asuka Seiyaku | | |
| Chemical compound, drug | hCG | Asuka Seiyaku | | |
| Chemical compound, drug | Indinavir | Sigma | SML0189 | 100 nM in water |
| Chemical compound, drug | Oligomycin | Sigma | O4876 | |
| Chemical compound, drug | 2-Deoxy-Glucose | Sigma | D8375 | |
| Chemical compound, drug | FCCP | Sigma | SML2959 | |
| Chemical compound, drug | Rotenone | Sigma | R8875 | |
| Chemical compound, drug | Antimycin A | Sigma | A8674 | |
| Chemical compound, drug | Lipid mixture | Sigma | L0288 | |
| Chemical compound, drug | Poly-L-lysine | Nacalai Tesque | 28356-84 | Dissolve poly-L-lysine in water to a concentration of 10 µg/ml, dispense 800 µl into a 12-well plate (2 µg/cm²), and incubate overnight at 37°C. |
| Chemical compound, drug | Collagen-coated plates | IWAKI | 4815-010 | |
| Chemical compound, drug | Epithelial Cell Medium | ScienCell | 4101 | |
| Software, algorithm | ImageJ | National Institutes of Health | Software version 2.1.0/1.53u, RRID:SCR_003070 | |
| Software, algorithm | CellSens imaging software | EVIDENT | Software version 4.1.1 | |
| Software, algorithm | Agilent Seahorse Wave software | Agilent Technologies Inc | Software version 3.0.0.41, RRID:SCR_019540 | |

*Appendix 1 Continued on next page*

*Appendix 1 Continued*

| Reagent type (species) or resource | Designation | Source or reference | Identifiers | Additional information |
|---|---|---|---|---|
| Software, algorithm | RaNA-seq pipeline (Fastp, Salmon, DESeq2) | ***Prieto and Barrios, 2019***, PMID: 31730197 | | |
| Software, algorithm | R | | Version 4.3.1, RRID:SCR_001905 | |
| Software, algorithm | CASA software | Hamilton Thorne | Software version 1.11.9 | |

