## [Editor Report · eLife Assessment]

This **important** work elucidates the biological processes and detailed mechanisms by which testosterone influences seminal plasma metabolites in mice. The evidence supporting the upregulation of metabolic enzymes and the role of ACLY is **solid**, highlighting the potential contributions of fatty acids to sperm motility.

---

## [Referee Report · Reviewer #1 (Public review)]

Summary:

In this revised report, Yamanaka and colleagues investigate a proposed mechanism by which testosterone modulates seminal plasma metabolites in mice. The authors have made improvements from the previous version by softening the claim that oleic acid derived from seminal vesicle epithelium strongly affects linear progressive motility in isolated cauda epididymal sperm in vitro. They have also addressed the ambiguous references to the strength of the relationship between fatty acids and sperm motility, making the manuscript more balanced and nuanced.

Strengths:

This study addresses an important gap in our understanding of how testosterone influences seminal plasma metabolites and, in turn, sperm motility. The findings provide valuable insights into the sensitivity of seminal vesicle epithelial cells to testosterone, which could improve in vitro conditions for studying sperm motility. The authors have added methodological details and re-performed experiments with more appropriate control groups, enhancing the robustness of the study. These revisions, along with more carefully modified language reflecting measurement nuances, add significant value to the field. The study's detailed exploration of the physiological role of reproductive tract glandular secretions in modulating sperm behaviors is likely to be of broad interest, providing a strong foundation for future research on the relationship between fatty acid beta-oxidation and sperm motility patterns.

Weaknesses:

While the connection between media fatty acids and sperm motility patterns is still not fully conclusive, the authors have taken substantial steps to clarify and tone down their conclusions. The revised manuscript presents a more balanced view, acknowledging the complexity of the relationship and providing a more solid basis for follow-on studies.

---

## [Referee Report · Reviewer #2 (Public review)]

Using a combination of in vivo studies with testosterone-inhibited and aged mice with lower testosterone levels, as well as isolated mouse and human seminal vesicle epithelial cells, the authors demonstrate that testosterone induces an increase in glucose uptake. The study reveals that testosterone triggers differential gene expression, particularly focusing on metabolic enzymes. They specifically identify increased expression of enzymes regulating cholesterol and fatty acid synthesis, leading to heightened production of 18:1 oleic acid. The revised version of the manuscript significantly strengthens the role of ACLY as a central regulator of seminal vesicle epithelial cell metabolic programming. The authors suggest that fatty acids secreted by seminal vesicle epithelial cells are taken up by sperm, resulting in a positive impact on sperm function. While the lipid mixture mimicking the lipids secreted by seminal vesicle epithelial cells shows marginal positive effect on sperm motility, the authors have made considerable progress in refining their conclusions. The revised manuscript acknowledges the complexity of pinpointing the specific seminal vesicle fluid component that potentially positively affects sperm function, providing a more measured and credible interpretation of their findings.

---

## [Author Response]

The following is the authors’ response to the previous reviews

**Editor's note:**
Thank you for taking time and efforts to improve this study. After re-review, two reviewers have a consensus that the connections the fatty acids and sperm motility is still ambiguous. Thus, I recommend to further tone down this conclusion consistently in the title and the text pointed out by reviewers before making a final version of record.

We sincerely appreciate the considerable time and effort you and the reviewers devoted to evaluating our manuscript. We have revised the title and text to express the relationship between fatty acids and sperm motility more consistently and toned down. With these revisions, we would like to proceed with publishing the manuscript as the Version of Record (VoR). Thank you very much for your guidance in improving our study.

**Public Reviews:**

**Reviewer #1 (Public review):**
Summary:In this revised report, Yamanaka and colleagues investigate a proposed mechanism by which testosterone modulates seminal plasma metabolites in mice. Based on limited evidence in previous versions of the report, the authors softened the claim that oleic acid derived from seminal vesicle epithelium strongly affects linear progressive motility in isolated cauda epididymal sperm in vitro. Though the report still contains somewhat ambiguous references to the strength of the relationship between fatty acids and sperm motility.Strengths:Often, reported epidydimal sperm from mice have lower percent progressive motility compared with sperm retrieved from the uterus or by comparison with human ejaculated sperm. The findings in this report may improve in vitro conditions to overcome this problem, as well as add important physiological context to the role of reproductive tract glandular secretions in modulating sperm behaviors. The strongest observations are related to the sensitivity of seminal vesicle epithelial cells to testosterone. The revisions include the addition of methodological detail, modified language to reflect the nuance of some of the measurements, as well as re-performed experiments with more appropriate control groups. The findings are likely to be of general interest to the field by providing context for follow-on studies regarding the relationship between fatty acid beta oxidation and sperm motility pattern.Weaknesses:The connection between media fatty acids and sperm motility pattern remains inconclusive.

We would like to express our sincere gratitude to the judges for their cooperation in reviewing the manuscript and for your helpful comments, which were instrumental in improving manuscript.

**Reviewer #2 (Public review):**
Using a combination of in vivo studies with testosterone-inhibited and aged mice with lower testosterone levels as well as isolated mouse and human seminal vesicle epithelial cells the authors show that testosterone induces an increase in glucose uptake. They find that testosterone induces a difference in gene expression with a focus on metabolic enzymes. Specifically, they identify increased expression of enzymes regulating cholesterol and fatty acid synthesis, leading to increased production of 18:1 oleic acid. The revised version strengthens the role of ACLY as the main regulator of seminal vesicle epithelial cell metabolic programming. The authors propose that fatty acids are secreted by seminal vesicle epithelial cells and are taken up by sperm, positively affecting sperm function. A lipid mixture mimicking the lipids secreted by seminal vesicle epithelial cells, however, only has a small and mostly non-significant effect on sperm motility, suggesting the authors were not apply to pinpoint the seminal vesicle fluid component that positively affects sperm function.

We greatly appreciate the reviewer’s thoughtful comments and time spent reviewing this manuscript. The relationship between lipids such as fatty acids and sperm motility remains unclear in the current dataset. Therefore, before finalizing the manuscript, we revised the title and text, as suggested by the reviewers, to express this conclusion more cautiously and consistently.

**Recommendations for the authors:**

**Reviewer #1 (Recommendations for the authors):**
Some additional comments are provided below to aid the authors in improving the quality of the work:Major Comments:(1) In the newly added supplemental figure 5, the authors note that the percentage data were arcisine transformed prior to statistical analysis without providing any other justification. This seems strange, especially for such a small sample size. It seems more appropriate for the authors to use a nonparametric test. Forcing symmetry without knowing what the shape of the true distribution is makes the ANOVA hard to interpret. Additionally, why use pairwise comparisons rather than comparing each group to the control (LM 0%). Also, note that the graphs are not individually labeled to distinguish them in the legend (A, B, C, etc.). Ultimately, the treatment differences don't seem that meaningful, even if the authors were able to observe statistical significance with the somewhat over-manipulated method of analysis.

Ultimately, the conclusion of this experiment (Supplemental figure 5) remains unchanged, but we agree that the relationship between fatty acids and sperm motility remains unclear. Therefore, before finalizing the manuscript, we revised the title and text as pointed out by the reviewers to express this conclusion more cautiously and consistently throughout the manuscript.

Arcsin transform is commonly used for percentage data [Zar, J.H. 2010. Biostatistical analysis., McDonald, J.H. 2014. Handbook of biological statistics.]. If the values are low or high, such as 0 to 30% or 70 to 100%, without arcsine transformation will result in a large deviation from the normality of the data. However, even if such a conversion is performed, it does not necessarily mean that the assumptions of normality and homogeneity of variance, which are prerequisites for parametric statistical analysis methods, are satisfied.

Given the small sample size and the possibility of non-normal data, we performed Shapiro–Wilk tests for each group (n = 6) and found no departure from normality (all p > 0.1). Q–Q plots and Levene’s test (p > 0.1) likewise supported the assumptions of ANOVA. Following the reviewer’s recommendation, we repeated the analysis with a Kruskal–Wallis test followed by Dunn’s post-hoc comparisons (Bonferroni corrected). Both approaches led to the same conclusions, with non-parametric p-values equal to or smaller than the parametric ones. In the revised manuscript we now report ANOVA as the primary analysis. The author response image includes effect sizes with 95 % confidence intervals, and provide the non-parametric results for transparency.

**Author response image 1. sa3fig1:** Results of reanalysis of supplementary Figure 5 using nonparametric tests and effect sizes with 95% confidence intervals. Upper part; Differences between groups were assessed by Kruskal–Wallis test, differences among values were analyzed by Dunn’s post-hoc comparisons (Bonferroni corrected) for multiple comparisons. Different letters represent significantly different groups. Lower part; The effect sizes with 95 % confidence intervals. For example, Cliff's Δ = -1 (95% CI ~ -0.6) in VSL's “LM 0 vs LM1” means that LM 1% values exceed LM 0 %values in all pairs.

(2) I appreciate that the authors toned down the interpretation of the effects of seminal plasma metabolites on sperm motility with a cautionary statement on Lines 397-405 and Line 259. However, they send mixed signals with the title of the report: "Testosterone-Induced Metabolic Changes in Seminal Vesicle Epithelial cells Alter Plasma Components to Enhance Sperm Motility", and on line 265 when the say "ACLY expression is upregulated by testosterone and is essential for the metabolic shift of seminal vesicle epithelial cells that mediates sperm linear motility".

The wording has been softened overall. The title has been changed to “Testosterone-Induced Metabolic Changes in Seminal Vesicle Epithelium Modify Seminal Plasma Components with Potential to Improve Sperm Motility” In the results (lines 265-266), we have stated that “ACLY expression is upregulated by testosterone and is essential for the metabolic shift that is associated with increased linear motility” without implying a causal relationship.

Minor Comments:(1) Typo on line 31: "understanding the male fertility mechanisms and may perspective for the development of potential biomarkers of male fertility and advance in the treatment of male infertility."

We have made the following corrections. “These findings suggest that testosterone-dependent lipid remodeling may contribute to sperm straight-line motility, and further functional verification is required.”

(2) Line 193: the statement is confusing "Therefore, we analyzed mitochondrial metabolism using a flux analyzer, predicting that more glucose is metabolized, pyruvate is metabolized from phosphoenolpyruvic acid through glycolysis in response to testosterone, and is further metabolized in the mitochondria." For example, 'Metabolized through glycolysis' is an ambiguous way to describe the pyruvate kinase reaction. Additionally, phosphoenolpyruvate has three acid ionizable groups, two of which have pKa's well below physiological pH, so phosphoenolpyruvate is the correct intermediate rather than phosphoenolpyruvic acid. The authors make similar mistakes with other organic acids such as citric acid.

Rewritten as “We therefore examined cellular energy metabolism with a flux analyzer, anticipating that testosterone would elevate glycolytic flux, thereby producing more pyruvate from phosphoenolpyruvate. Because extracellular pyruvate levels simultaneously declined, we inferred that the cells had an increased pyruvate demand and, at that time, hypothesized that the excess pyruvate would enter the mitochondria to support enhanced oxidative metabolism.” (lines 193-198)

The organic acids are now referenced in their appropriate forms (e.g., citrate, phosphoenolpyruvate).

(3) Line: 271: "Acly" should be all capitalized to "ACLY". The report mixes capitalizing through out and could be more consistent.

We appreciate the reviewer’s attention to nomenclature and have standardized the manuscript accordingly. Proteins are written in Roman letters, all in capital letters. Mouse gene symbols: italics, first letter capitalize.

**Reviewer #2 (Recommendations for the authors):**
Major comments:(1) 'Once capacitation is complete, sperm cannot maintain that state for a long time'. The publications cited by the author do not support that statement and this reviewer also does not agree. Lower fertilization efficiency from in vitro capacitated epidydimal sperm does not have to mean capacitation is reversed, it can simply mean in vitro capacitation conditions not accurately mimic capacitation in vivo.

We thank the reviewer for pointing this out and would like to clarify our position. Our statement does not suggest a "reversal" of active capacitation. Rather, it reflects the well-documented fact that capacitation is a transient process. Sperm that undergo capacitation too early cannot maintain that state for long enough to retain their ability to fertilize at the moment and location of fertilization in vivo.

(2) How do the authors explain the discrepancy between the results shown in Fig. S1E, the increase in sperm motility upon mixing of sperm with SVF and the results reported in Li et al 2025. Mentioning decapacitating factors without further explanation is insufficient.

We appreciate the reviewer's feedback pointing out the need for a clearer explanation.

Seminal plasma is inherently binary, containing both decapacitation factors that delay or inhibit capacitation and nutrient substrates that promote sperm motility.

In vivo, it is believed that the coating of sperm by decapacitation factors is removed by uterine fluid and albumin as it passes through the female reproductive tract [PMID: 22827391, PMID: 24274412]. In contrast, standard fertilization culture media lack a clearance pathway, so decapacitating factors are retained throughout the culture period. As a result, the cleavage rate after in vitro fertilization using sperm exposed to seminal vesicle fluid decreased dramatically.

Lipids, such as fatty acids, increased sperm motility without directly inducing markers of fertilization. These results suggest that the enhancement of motility by lipids is functionally distinct from the capacitation-inhibiting function of seminal plasma proteins. The data from this study are consistent with the biphasic model. Specifically, decapacitation factors temporarily stabilize the sperm membrane, preventing early capacitation. Meanwhile, lipids enhance sperm motility, enabling them to rapidly pass through the hostile uterine environment.

(3) This reviewer does not see the merit in including a lipid mixture motility experiment compared to using OA alone. The increase in motility is still small and far from comparable to the motility increase with seminal vesicle fluid. In this reviewer's opinion the experiment is still inconclusive and should not be highlighted in the manuscript title.

The wording has been softened overall. The title has been changed to “Testosterone-Induced Metabolic Changes in Seminal Vesicle Epithelium Modify Seminal Plasma Components with Potential to Improve Sperm Motility”. (Please see also Reviewer 1's main comment 1)

Minor comments:(1) 'This change includes a large amplitude of flagella' does not make sense. Please correct.

The following corrections have been made. “This change is characterized by large-amplitude flagellar beating.” (lines 44-45)